# In-chip critical plasma seeds for laser writing of reconfigurable silicon photonics systems

Andong Wang [1,2], Amlan Das [1,7], Vladimir Yu Fedorov [3,4], Pol Sopeña [1], Stelios Tzortzakis [3,5,6] & David Grojo [1] ✉

Ultrafast laser three-dimensional writing has made breakthroughs in manufacturing technologies. However, it remains rarely adopted for semiconductor technologies due to in-chip propagation nonlinearities causing a lack of controllability for intense infrared light. To solve this problem, plasma-optics concepts are promising since ultrashort laser pulses, even if inappropriate for direct writing, can readily inject high-density free-carriers inside semiconductors. To achieve highly localized and reliable processing, we create plasma seeds with tightly focused pre-ionizing femtosecond pulses. We show how critical density conditions can be used for extremely confined energy deposition with a synchronized writing irradiation and create ~1-μm-sized isotropic modifications inside silicon. Drastic improvement is also found on the material change controllability leading to unique demonstrations including rewritable optical memories (>100 writing/erasure cycles) and graded-index functionalities. By solving its controllability issues with critical plasma seeds, we show the potential of ultrafast laser writing for flexible fabrication of reconfigurable monolithic silicon-based optical devices.

Silicon remains the basis materials in microelectronics and integrated photonics. Capitalizing on the huge investments made in silicon foundries and continuously growing innovations, a huge panel of extremely precise and highly productive micro-/nano-fabrication solutions are today available. However, it is striking to note that these remain primarily based on planar (e.g., lithography) or surface structuring technologies (e.g., ion beams). Multi-steps and tedious production cycles are then required to deliver the three-dimensional (3D) architecture products. Moreover, despite their extreme level of miniaturization, the ultrahigh density devices are developed on silicon wafers which, even if getting thinner and thinner, leaves a wide unexploited space under the chips. Inspired by the development over the last few decades of advanced femtosecond laser writing technologies as for instance super-resolution 3D printing[1,2], perennial optical memory technologies[3,4] or direct formation of microscale 3D optical

elements inside transparent dielectric materials[5,6], recent researches attempt to translate the approach in semiconductors using intense infrared light. However, a more challenging situation is reported due to the inherent properties of narrow gap materials. In particular, the high refractive index and strong optical nonlinearities prevent to achieve a sufficient space-time energy localization to 3D write semiconductors with the shortest pulses or those typically used for precise processing in dielectrics[7-10]. It is only recently that demonstrations have emerged using non-conventional configurations (e.g., solid-immersion hyperfocusing) and/or appropriate combinations of longer pulses[11-15]. In this context, a major advance has been made by O. Tokel's group[16] on the question of process resolution, a key aspect to meet the requirement for advanced microelectronics applications. However, despite the impressive nanometre precision demonstrated in this work, a drawback of the method is the required preparation of

[1]Aix Marseille University, CNRS, LP3 UMR7341, Marseille, France. [2]Laser Micro/Nano Fabrication Laboratory, School of Mechanical Engineering, Beijing Institute of Technology, Beijing, China. [3]Science Program, Texas A&M University at Qatar, Doha, Qatar. [4]P. N. Lebedev Physical Institute of the Russian Academy of Sciences, Moscow, Russia. [5]Institute of Electronic Structure and Laser (IESL), Foundation for Research and Technology—Hellas (FORTH), Heraklion, Greece. [6]Materials Science and Technology Department, University of Crete, Heraklion, Greece. [7]Present address: Currently at Centre for Functional Materials, Vellore Institute of Technology, Vellore, Tamil Nadu, India. ✉e-mail: david.grojo@cnrs.fr

pre-modified structures for writing at this level of performance[16]. While these works constitute interesting proof-of-concepts supporting the feasibility of 3D laser fabrication in silicon with excellent precision, all applied methodologies have in common that they remain hardly applicable for real device fabrication.

For a digital writing solution, we describe here a unique microplasma-based solution to this problem, which can be implemented in a conventional 3D machining setup with a simple dual-beam configuration. The introduced general concept translates for the first time the highly deterministic and high-resolution direct writing capabilities demonstrated with ultrafast laser on surfaces[17] to the bulk. It also leads to point defect creation and local phase changes in crystalline silicon for unique refractive index engineering capabilities as well as new reconfigurability features, which are inaccessible otherwise.

## Results

### Pre-ionization for improved writing

Our solution is inspired by the emerging concept of "plasma optics" exploiting the transient optical properties of plasmas for light manipulation. Advanced demonstrations are various and include transient mirrors[18], holograms[19] or reconfigurable functions in integrated circuits[20] but all rely on near-critical surface plasmas. In this work, we target similar plasmas inside the bulk of silicon to in situ influence a processing beam. In previous works, we already observed that relatively dense plasmas could be generated inside the bulk of silicon with ultrafast laser pulses when tight focusing conditions are used[11,21]. While this was not associated with permanent modifications, the locally obtained strong permittivity change is expected to potentially influence any further incoming light and reshape energy deposition, an aspect which constitutes the basis of the proposed solution and illustrated in Fig. 1a.

Before analysing the details of the material response with appropriate dual-beam conditions, we first highlight the pre-requisite of this method: achieving critical plasma density conditions under breakdown threshold. This condition is rather uncommon in solids and not reported in dielectrics but interestingly achievable in semiconductors because the absorbed energy to promote electrons in the conduction band directly scales with the band gap value. The most direct way to reveal such conditions inside silicon is to rely on ultrafast infrared microscopy. Using a specific pump-probe configuration with hyper-focused beams (see Methods section), we access details of the produced microplasmas right after irradiations with <200-fs, 1550-nm pulses (NA = 0.85) at different energy levels. Examples of the obtained transmission images are shown in Fig. 1b, from which we extract the minimum local transmission found at the plasma fronts (facing incoming light). The corresponding free-carrier density is calculated from the inverse Bremsstrahlung absorption described with a simple Drude model[9], and is presented in Fig. 1b as a function of input energy. Above a pulse energy of $\approx$100 nJ, the measurements indicate near-zero transmission and a free carrier density $>3.0 \times 10^{20}$ cm$^{-3}$ that corresponds well with the theoretical critical plasma density at the applied laser wavelength ($N_c \approx 4.6 \times 10^{20}$ cm$^{-3}$ at 1.55 μm). Considering the energy used to create these strong ionization conditions, one can directly make a lower estimate by simply multiplying this electron density by the band gap of Si (1.1 eV). This leads to a very modest energy density of ~50 J cm$^{-3}$, far below material melting conditions[11] for Si (typ. 5 kJ.cm$^{-3}$). This describes a situation without equivalent in dielectric studies. Wider band gaps (typ. 5–10 eV) and higher critical densities due to the most common use of shorter wavelengths ($N_c > 10^{21}$ cm$^{-3}$ for visible wavelengths) generally cause material breakdown conditions well under those to create plasmas with similar optical properties.

After establishing the critical plasma seeds at modest energy density using femtosecond pulses in semiconductors, it is important to

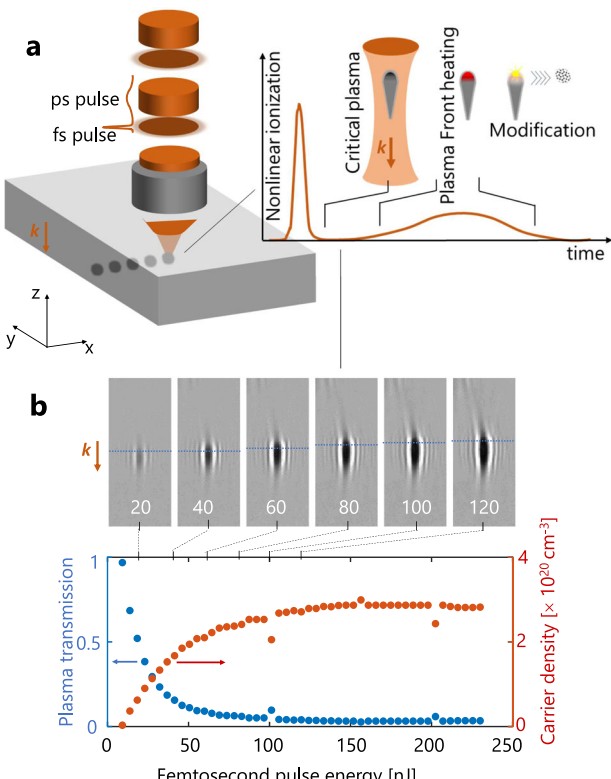

**Fig. 1 | Critical density plasmas for localized modification writing. a** Illustration of the critical-plasma seeding concept using synchronized femtosecond and picosecond pulses to achieve highly localized internal structuring of silicon. **b** Ultrafast infrared microscopy images of microplasmas produced inside Si with 200-fs pulses focused at NA = 0.85 (snapshots at 20-ps delay) and measured minimum transmission levels in the plasma images as function of pulse energy. The correspondingly peak free-carrier densities (calculated using a Drude model) are always found on the plasma front (dashed line on the images) and reach values approaching the critical plasma density ($N_c \approx 4.6 \times 10^{20}$ cm$^{-3}$).

note in Fig. 1b a saturation behaviour. In practice, when the energy is increased, the absorption front and plasma gradually develop in the pre-focal zone without any net increase of the peak plasma density. Then, the only way to deposit more energy and achieve permanent modifications is to deliver and manipulate laser radiation over longer time-scales. This is consistent with recent studies in single pulse configurations revealing that internal structuring inside Si requires a sufficiently long pulses to create, heat up electrons and deposit enough energy for material breakdown (typically >5 ps with NA = 0.85)[22]. However, limited by the rise of heat-driven effects, the writing process exhibits in this case a modest writing resolution significantly above the diffraction limit. As shown with Fig. 2a, with 10-ps pulses the best writing resolution is >6 μm laterally for an expected optical spot size ~λ/2NA = 0.91 μm at Abbe's limit, and >12 μm longitudinally for a confocal parameter of ~4 μm. This writing performance obviously cannot meet the high-precision requirements of most semiconductor applications.

By creating an appropriate plasma seed, a remarkable finding of this work is that the resolution of the same writing pulse can be drastically improved. This is immediately seen with the comparative images in Fig. 2a, adding an appropriate femtosecond pre-pulse 20 ps before the applied picosecond pulse. With the double-pulse configuration, one first notices a decrease of the apparent modification threshold which is similar to an observation made in a previous work using ultrashort pulses exhibiting a degraded contrast[15]. However, we also add here the capacity of producing significantly smaller written spots for conditions between the writing thresholds

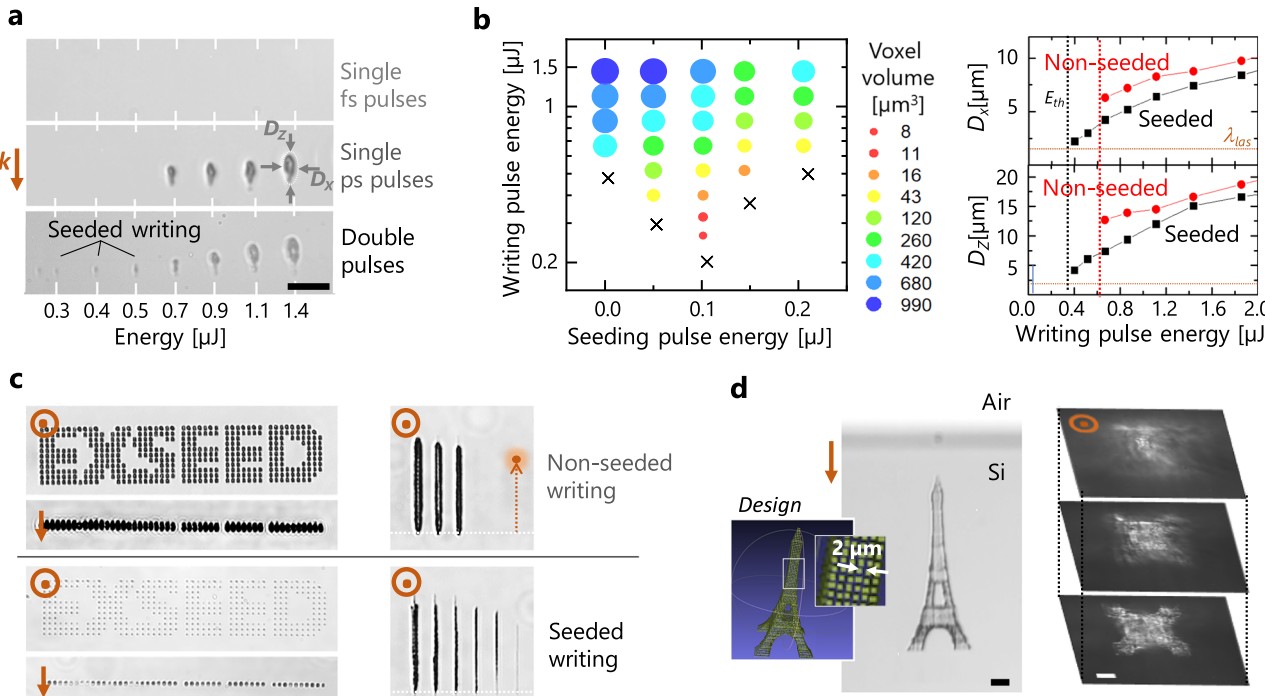

**Fig. 2 | Precision 3D writing in silicon using critical plasma seeds. a** Infrared microscopy observations of modifications (lateral view) made inside silicon with varying energies by using 200-fs pulses (no modification), 10-ps pulses (non-seeded writing) and double-pulse sequences (seeded writing) composed of a 200-fs pre-pulse of 0.14-μJ energy and a 10-ps pulse (energy indicated below) at a 20-ps delay. The irradiations are performed with 0.85NA focusing conditions at 1550-nm wavelength. The number of applied pulses is 100 for each irradiation. At low energy smaller additional features are produced with the seeded writing configuration. **b** Volume of the written voxels as a function of the energy for each applied pulse in the double-pulse sequences and comparison of the lateral and longitudinal writing resolutions with and without the use of a femtosecond pre-pulse of 0.14-μJ energy.

Plasma seeding with double-pulse irradiation leads to decreased writing thresholds and smaller features. **c** Demonstration of improved writing performances (digital and line scanning) by critical-plasma seeding. Dots are produced applying 100 pulses. The writing speed for the lines written at different energy levels is 100 μm/s and the repetition rate 1 kHz. **d** 3D Eiffel tower written below the surface of a silicon wafer with a precision level only accessible by plasma-seeded writing. Observations are made using lateral transmission and top-view dark field infrared microscopy. The structure is designed on the basis of written voxels separated by 2 μm. Each voxel is written using optimum double-pulse irradiations as identified in b. Scale bars are 20 μm.

for the seeded and non-seeded writing cases (bottom image of Fig. 2a).

To determine the smallest achievable writing volume with plasma seeds, we produce modifications with different energies for the applied femtosecond and picosecond pulses. As shown with Fig. 2b, in comparison to the modified volumes obtained with pure picosecond irradiation (first column), a "sweet" valley is created above the reduced writing threshold (black crosses are tested conditions under the writing threshold) when adding a femtosecond pre-pulse of about 0.1-μJ energy. Nearly 50 times improvement of the minimum modified volume is found (see red spots). Concentrating on the use of this best identified conditions for pre-ionization (0.14-μJ fs pulse energy), we use the set of collected infrared images to also report in Fig. 2b on the lateral and longitudinal sizes of the produced features. Interestingly, it shows a lateral writing resolution close to the laser wavelength (<1.6 μm) and corresponding to an improvement exceeding 35% in comparison to single pulse writing. An even more impressive improvement >75% is also observed for the longitudinal precision (modifications down to 2.6 μm) largely overcoming the confocal parameter of the beam for diffraction limited conditions. Moreover, it is worth noting with these measurements that the use of the critical plasma seed concept offers the capacity to reach this superior performance in a comfortable processing window above the threshold condition ($E_{th} < E < 1.6\, E_{th}$).

To prove the reliability of the approach for super-resolution writing, we produced 2D arrays of modifications. For fair comparisons between seeded and non-seed cases, the applied writing pulse energy

is set 10% above the writing threshold (1.1 $E_{th}$) in both cases (best resolution for conditions not affected by repeatability limitations caused by <1% pulse-to-pulse energy fluctuation of the laser). As it is shown in Fig. 2c, repeatable writing is achievable in both cases for these conditions. An analysis of the images gives a spot size variability <12% (relative standard deviation) for both approaches. However, the superior writing resolution with femtosecond pre-ionization is confirmed. Important for technological considerations, this precision holds for static irradiation but also for scan writing configurations (longitudinal and transverse) as evidenced by comparative modified lines also shown in Fig. 2c. The benefit is also independent of the processing depth provided that beams compensated for spherical aberration are used. This is demonstrated with Fig. 2d showing the 3D writing of a < 200-μm Eiffel tower under the surface of a silicon wafer by a 3D scanning procedure including a readjustment of the aberration compensation at each of the three levels of the tower (see details of the writing procedure in Supplementary Note 2). This corresponds to a realization requiring a level of precision inaccessible by picosecond laser writing in semiconductors (See additional performance comparisons in Supplementary Note 2) and flexibility out-of-reach for the other advanced single-beam schemes[11,16].

## Critical plasma seeds as drivers

To reveal the underlying mechanisms behind the improved resolution performance, we repeat the plasma imaging study with the two-pulse irradiation. Figure 3a-A shows the plasma image obtained with the ideal pre-pulse only. With repeated irradiation with the two-pulse

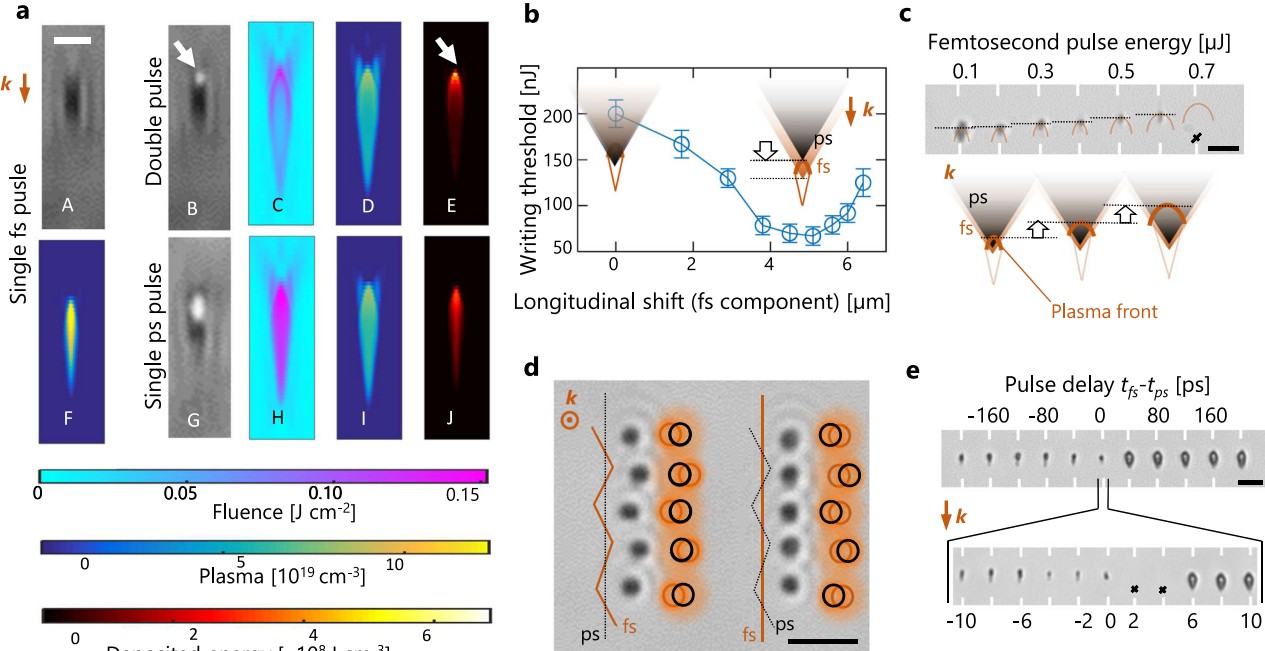

**Fig. 3 | Simulation and experimental evidences of the driving role of critical plasma fronts produced by femtosecond pre-pulses. a** Comparisons between ultrafast infrared microscopy images of produced microplasmas (grey background), and calculated distributions across the focus for the laser fluence (cyan background), the plasma density (blue background), and energy absorption (black background). The comparison includes the response to each individual pulse and the combined configuration for improved writing. The white arrows show the good correspondence between the breakdown and absorbed energy localization on the microplasma front. The characteristics of applied femtosecond and picosecond pulses are described in Methods sections. The scale bar is 5 μm. **b**–**e** Material modification responses when the relative position in time and space of the two focused pulses is varied. **b** The relative change of the longitudinal spot positions reveals a minimum energy threshold for writing at Δz≈-5 μm (z = 0 corresponds to

the best overlap). This corresponds well with the difference between the geometrical focus of the femtosecond laser spot and the plasma front position as can be seen in a. **c** As expected for the plasma front, the modifications move backward when increasing the femtosecond pulse energy and the picosecond pulse energy is maintained at 0.7 μJ. **d** The relative change of lateral positions of the spots reveal modification always following the seeding spot (fs component) and not the picosecond writing component. **e** When the delay between the two pulses is varied, the performances of the seeded and non-seeded writing configurations are confirmed with the observations for negative and positive delays, respectively. When pulses overlap, one observes an inhibition of the writing at specific delays (see black crosses) attributed to the energy deposition, which is split in distinct volumes. Scale bars are 20 μm.

sequence, the image Fig. 3a-B exhibits a bright spot as soon as breakdown is initiated. This directly locates a centre for preferential energy deposition and modification at the plasma front. Due to resolution and sensitivity limits that do not allow to reveal more details of the plasma distributions, we turned rapidly to nonlinear propagation simulations to interpret the observations. We use the approach described in ref. 23. to derive nonparaxial input conditions for simulations of tightly focused pulses (see description in Methods section). First, we try to understand from the simulations how the plasma influences the energy flux propagation. We concentrate here the discussion on the observables near focus, but previous works[8–10] have revealed that the created low-density plasma all along the beam path can reshape energy flux in semiconductors. In most cases, the effect of the pulses that create the plasmas is usually a depletion and delocalization of the energy flux but would correspond here to a collapse at the specific pre-focal point for our writing pulse. To examine this hypothesis, we show in Fig. 3a-C, a-H the calculated fluence distributions (cross-section along propagation axis) for applied picosecond pulses with and without pre-ionization. This and more detailed analyses given in Supplementary Note 4 evidence conditions in which the energy flux is not reshaped by pre-ionization until the plasma front is reached. However, the energy flux is stopped at the plasma front, as clearly shown by the shell-like distribution in Fig. 3a-C. This is a direct consequence of the critical plasma density conditions. Using a simple Drude model, we can predict (not shown) an optical penetration depth <1 μm before the rise of plasma reflectivity and so important losses for a density >1 × 10²⁰ cm⁻³. This compares remarkably well with the shell thickness

visible in the two-pulse fluence maps (Fig. 3a-C) for which similar peak plasma density is coincidently expected from plasma observations (Fig. 3a-B).

When simulating the plasma generation, the peak of plasma density is also found on its front as can be seen with Fig. 3a-D. However, a relatively modest contrast is obtained with respect to the rest of the plasma (linear colour scale). We can also notice that the net result of the plasma decay between pulses and the modest ionization yield of the additional picosecond pulse actually lead to a peak plasma density (Fig. 3a-D, I) inferior to the pure femtosecond pulse (Fig. 3a-F). It is only when we represent the deposited energy density accounting for Joule heating of the plasma by the picosecond component (Fig. 3a-E, J) that a much more contrasted sub-microscale region corresponding to the permanent modification zone is obtained. The conclusion is then that the plasma front effectively assists the laser energy deposition inside the bulk in much the same way one can also guide lightning electrical breakdown along plasma filaments in the atmosphere[24]. This also explains well the reduced threshold (Fig. 2a, b) associated with resolution enhancement for the plasma seeded case.

To confirm the driving role of the created plasma front in these observations, we performed a series of experiments in which we varied in time and space the relative positions of the pre-ionizing pulse and the picosecond heating pulse. Because of the expected mismatch between the heating pulse focus and the pre-focal plasma front, we expect a process with efficiency that can depend on relative motion of the two beams along the optical axis. This is confirmed in Fig. 3b which shows an additional reduction of the modification threshold from 220

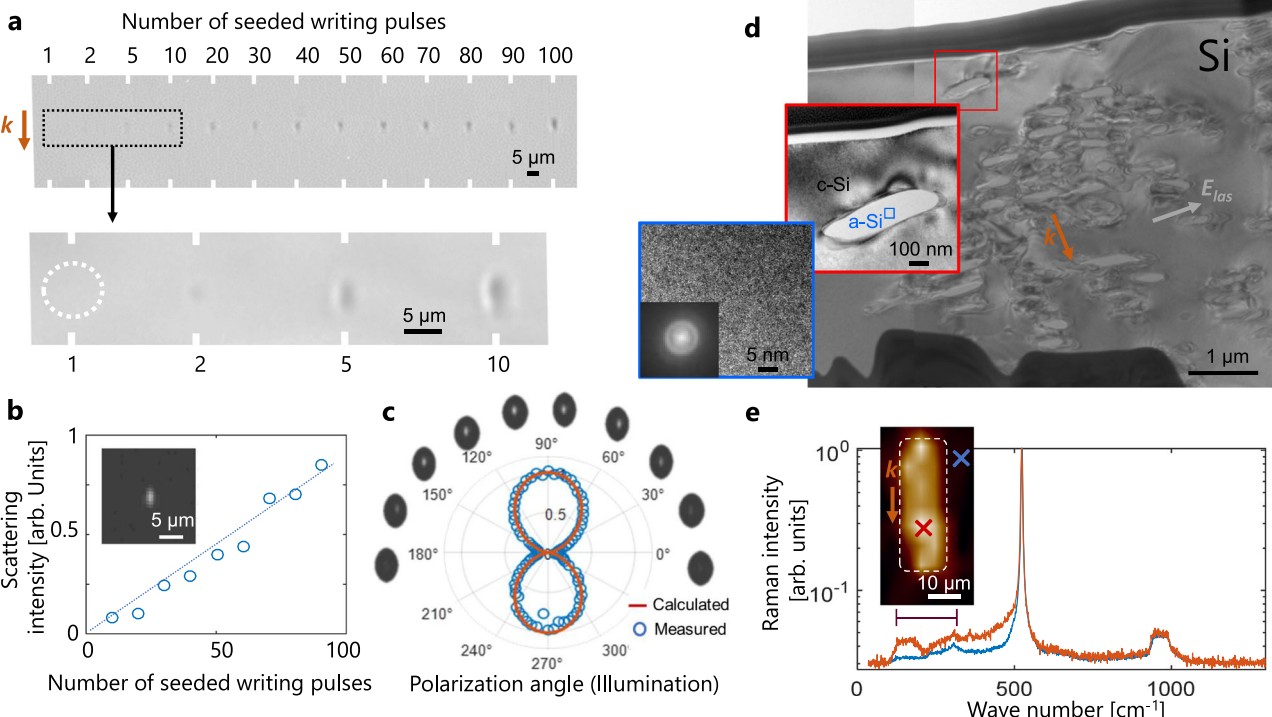

**Fig. 4 | Characterization of the produced modifications. a** Lateral infrared microscopy images of modifications produced with different number of applied double-pulses (0.14-μJ 200-fs seeding pulses synchronized with 0.4-μJ 10-ps pulses, 20-ps delay). **b** Scattering microscopy measurement using the lateral infrared microscope and illuminating the modification using CW light at 1550 nm along the same optical axis as for writing. The inset shows the dark field image of the voxel obtained with single double-pulse irradiation. The scattering intensity increases linearly as a function of the number of applied double-pulses for writing. **c** Polar plot of the scattering signal as a function of illumination polarization revealing an anisotropy in the modifications. **d** Transmission electron microcopy and **e** Raman micro-spectroscopy analyses for a cross-section of a large-volume modification obtained by a scanning procedure with the seeded writing beam. The observations reveal an assembly of amorphous nanovolumes in the modified zone.

nJ down to only 67 nJ for a focus shift $Z_{fs}$-$Z_{ps}$ ≈ 5 μm by changing the divergence of the picosecond beam. This distance corresponds well with the observed difference between the plasma ignition spot (geometrical focus) and the plasma front for the identified best seeding conditions (see Fig. 3a). Another way to move the critical seed on the optical axis is to change the applied pulse energy as the plasma front is expected to move backward (in the direction of the incoming radiation) at higher pulse energies. This is confirmed by experiments in Fig. 3c and simulations (see Supplementary Note 4). Interestingly, we can also similarly shift the plasma laterally by changing beam pointing directions before the focusing optics (details in Supplementary Note 5). As shown in Fig. 3d, we observe in this case that the modifications shift together with the femtosecond laser spot, while absolutely no change is found when the picosecond laser spot is moved by the same amount. Taken together, these spatial considerations prove the driving role of the plasma front enhancing conditions for writing but also inhibiting the interaction everywhere else (in particular behind plasma front) to improve modification localization.

The inhibition capabilities caused by the critical plasma front is confirmed when the temporal delay between the pulses is varied. In Fig. 3e, we show the modification results at different delays $t_{fs}$-$t_{ps}$ starting from −200 ps (femtosecond ionization first) to +200 ps (picosecond writing first). The drastic size difference between features obtained with positive and negative delays directly confirms the benefits from the plasma seeding effect. We also interestingly notice a total absence of permanent modification for a range of delays in which the two pulses temporally overlap (from ≈0 to ≈−4 ps). This is very consistent with the strong effect of the plasma front, which tends to split the energy deposition of the picosecond pulse in two distinct volumes. When the two pulses overlap, the first part of the picosecond

pulse deposits the energy in a large non-seeded volume (typically the focal volume of the picosecond beam corresponding to the modified zone without pre-ionization). For the remaining part of the pulse, the interaction is then transferred in the volume defined by the plasma front caused by seeding. This splitting in two interaction regions results in reduced deposited energy densities that fall below the modification threshold in the present cases.

## Unique material modification features

Given the high degree of ionization and the strongly nonequilibrium conditions expected when writing with critical plasma seeds, one can also expect access to new material modification characteristics. This is immediately confirmed by observations of the evolution of the modifications (side view of modified spots) under repeated irradiations. As shown in Fig. 4a, the produced features become rapidly nearly-independent to the number of applied pulses (typ. above 30 applied pulses). This indicates a very different response as one usually observes an important growth of the modifications with the number of applied pulses in previous experiments in semiconductors[25] but also in the various and numerous experimental studies performed in dielectrics[26]. This is attributed to the unique level of confinement and local contrast of the energy deposition with critical plasma seeds. Interestingly, the images of Fig. 4a also reveal nearly invisible modifications when produced by single and few-shot irradiation and a localization likely under the resolution limit of our infrared microscopy setup. However, the deterministic occurrence of defect creation can be systematically confirmed by scattering detection using a strong lateral illumination to create dark field images as shown in the insert of Fig. 4b.

To discuss the nature of the modifications, it is interesting to notice that the scattering intensity is proportional to the number of

applied writing pulses, as shown in Fig. 4b. This supports the creation of low-density scatterers accumulating on a pulse-to-pulse basis and thus with a density which can be precisely controlled. To further examine the properties of the produced defects, we measured the polarization-orientation dependence (for illumination) of the observed scattered light in this configuration. A strong anisotropy is revealed by the scattering images and the polar plot of the integrated scattering intensity presented in Fig. 4c. The latter matches remarkedly well a theoretical Rayleigh scattering response (red line). This evidences the formation of nanoscale or structural defects. This type of modification defers significantly to the previously written structures in longer, less intense but necessarily more energetic pulse regimes causing in most cases micro-scale nonuniformity and/or material disruptions[12,27].

We also found that the scattering signal is not only dependent to the polarization of the probing light but also to that of the double-pulse writing laser. To illustrate this, we write structures using two orthogonal polarizations and probe their scattering properties under the same measurement conditions (not shown). The ratios of the obtained scattering signals between the two writing configurations indicate almost no difference for the spots written with less than 10 pulses. However, a difference is progressively growing with more applied pulses. With 1000 applied pulses, the ratio reaches 2:1 which demonstrates a clear anisotropy in the produced nanostructures, starting from uniformly distributed point defects created in the focal volume which accumulate on a pulse-to-pulse basis (Fig. 4b) and create progressively a field-dependent anisotropy in the medium. By analogy with the low-loss structuring recently reported by femtosecond laser writing inside silica[28], the obtained anisotropy found in silicon must be also associated to birefringence. However, all our attempts to reveal it by cross-polarized transmission observations have been unsuccessful. This indicates a much weaker birefringence level and material modifications that deviates clearly from the formation of the nanopores as in silica.

To investigate these material modifications, we conducted transmission electron microscopy analyses of the modified silicon (Fig. 4d). Due to the complexity of sample preparation to analyse the extremely local structures ($\approx 1\,\mu m^3$) and even more so those which are optically *invisible*, we apply here scanning procedures to write sufficiently large volumes and extract a lamella intercepting the modified volume by focused ion-beam machining from the surface. According to irradiation parameters (see Supplementary Note 6), the effective number of applied pulses for the modification is >100 in the studied case. This corresponds to modifications which are optically well contrasted (see Fig. 4a) and exhibiting scattering anisotropy but are not necessarily relevant to reveal the initial local defects associated with the low-loss structures at low number of applied shots. While most TEM analyses in previous works concluded on a modified silicon region that remains mostly in its monocrystalline state (recrystallized and/or stressed), we made with our double-pulse configuration two unprecedented observations consistent with the measured anisotropic scattering responses. A first important observation is a significant proportion of amorphous silicon in small domains (typ. size about 100 nm) over the processed volumes. This is confirmed by the complementary Raman spectroscopy analyses shown in Fig. 4e which clearly reveals the distribution of the amorphous phase across the processed zone. The amorphization inside silicon is a long-pursued objective because amorphous silicon exhibits a significantly larger refractive index than crystalline silicon ($((n_a\text{-}n_c)/n_c \approx 7\%)$[29] and so it holds promises for photonics engineering. Various types of internal modifications in different regimes have been the subject of TEM and electron diffraction analyses. A partial amorphization was searched in most cases but the studies (including ours) observed mainly a stressed monocrystalline material. Evidence of amorphization was interestingly also found in the recent work of Trinh et al. in the nanosecond

regime[30]. However, it was only very residual with <0.2% of material transformed into amorphous state. Here, we do not show a spatial controllability of the phase change or the complete amorphization of written voxels that would immediately allow the fabrication of pure phase-change structures. However, we report for the first time on a significant amount of amorphous silicon in the processes zone. This indicates that the conditions for amorphization of silicon widely reported in surface studies with ultrashort pulses[29], are also likely achievable in the bulk. This opens the future vision of phase-change reversible silicon 3D devices, a promising concept envisioned to date only for 2D platforms[30]. The second important observation is amorphized domains taking the form of self-organized periodical structures consisting of multiple strips (Fig. 4d and more details in Supplementary Note 6). The periodical lines remain imperfect but the direction is parallel to the polarization direction, and the period is around 280 ($\pm 50$) nm. Femtosecond laser-induced well-defined nanogratings[3] correspond to another reported feature subsequent to nanopore formation for silica[28]. While the nature of material modifications clearly differs here, as for dielectrics, the spontaneous formation of nanogratings under repeated irradiation can be attributed to field-enhancement by interactions with pre-existing defects and non-uniformities progressively imposing the geometry for the obtained structuring inside silicon[31].

## 3D writing of reconfigurable silicon photonics functionalities

Reconfigurable technologies represent another long-pursued objective for many advanced applications, such as programmable/active photonic devices[32–35] or data storage[3,36,37]. Here, we found not only an enhancement of the controllability for the writing process but also a new type of structures that permit near-perfect local erasure and rewriting, a capability that we could not obtain without critical plasma seeds. The feasibility of erasure of laser-written structures in silicon by thermal treatment was first demonstrated in a work by Tokel et al.[10,12] which represents a strong basis for our reconfigurability demonstrations. First, we confirmed similar capabilities for the modifications produced in this work by tentative thermal erasure using a furnace. Using identical conditions for infrared observations, Fig. 5a shows features which vanish very well under thermal treatment for written modifications with femtosecond pre-ionization. For pure picosecond writing, recrystallisation and erasure are almost inevitably accompanied with crack/defect propagations attributed to the presence of material disruptions in the originally modified volumes. This makes a major difference with critical plasma seed writing, which relies on point defect formation in silicon (for low-loss structures before formation of amorphous grains) which can be efficiently suppressed by sub-melt thermal annealing according to surface studies[38].

On the basis of this interesting response, an important advance for reconfigurability demonstrations was the use of long laser pulses to achieve similar thermal erasure by localized treatment. According to results shown in Fig. 5b, we can then use the same objective lens as the one for ultrafast seeded writing to focus a nanosecond laser beam so that we can target the same zones for writing and erasure of voxels anywhere in 3D space inside the silicon samples. The required conditions for efficient laser erasure are detailed in Supplementary Note 7. To demonstrate the reliability of the process, Fig.5c shows the writing/rewriting of QR codes confined in a micrometre layer inside the bulk of a silicon wafer. The three images correspond to the same region where each spot is successively erased and/or rewritten so that new QR codes are created. While we can notice imperfections in the erased spots after three cycles for these conditions, the successful demonstration can be assessed by the reader by using any standard QR code scanner applied to the infrared microscopy images provided in Fig. 5c. We believe that this unique silicon in-chip or in-wafer information storage (and so non-invasive for surface circuits) is potentially attractive for counterfeiting or traceability measures in the semiconductor industry.

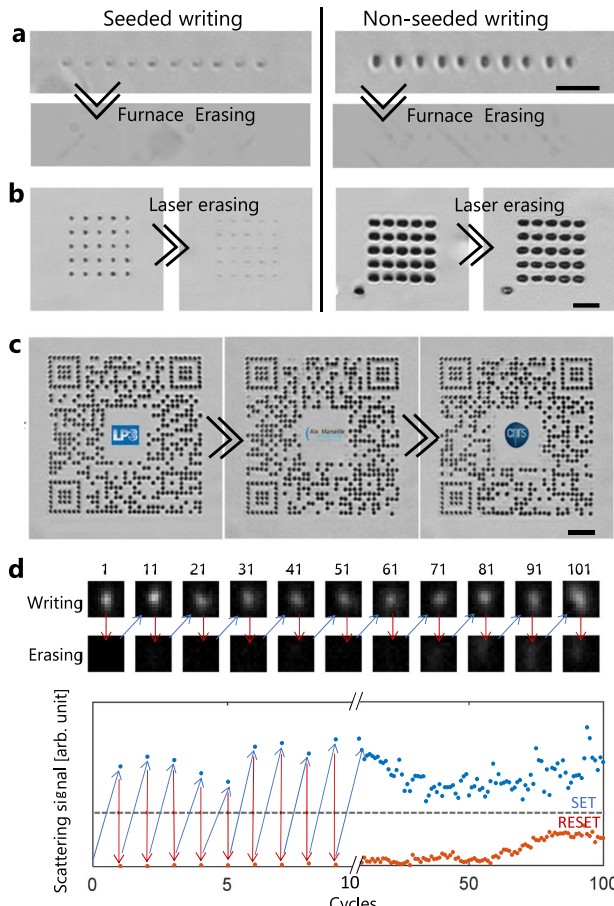

**Fig. 5 | Writing, erasing and rewriting inside silicon. a** Written voxels (infrared microscopy, lateral view) before (first row) and after (second row) thermal treatments using a furnace (>1000 °C for 4 h). The comparison reveals a superior erasing capacity when modifications are produced with the seeded writing approach (double pulse). **b** Similar voxels produced by the two writing methods (top view observations) before and after subsequent irradiation using a focused nanosecond laser (0.3µJ 5-ns 1550-nm pulses). Similar conclusions can be stressed with this scheme giving a solution for local erasure. **c** Three QR codes subsequently produced at the same location inside a silicon wafer (300-µm depth) by applying seeded writing, nanosecond laser erasing and seeded re-writing procedures. **d** Scattering images of a voxel repeatedly written and erased for more than 100 cycles with optimized double-pulse and nanosecond laser conditions. The graph represents the integrated scattering signal from the darkfield microcopy images acquired after each irradiation (examples shown on top). Scale bars are 20 µm.

To show the possibility to make much more cycles with engineering optimizations, we performed a study on a single spot under repeated writing/erasure irradiations so that we suppress any potential repositioning issues that may contribute to process imperfections. The microscopy images and corresponding scattering measurements shown in Fig. 5d directly evidence an excellent robustness for more than 100 cycles. By setting a given scattering threshold (shown by the dashed line in graph of Fig. 5d), we can readily define and differentiate some SET and RESET states supporting a general vision for enduring non-volatile optical memory applications.

To move from binary information encoding to refractive index engineering, we rely on the observation that the local defects created by our method can be gradually accumulated changing laser parameters (see Fig. 4b). This must translate into gradual changes of apparent optical properties opening also the door to reconfigurable and tunable photonic systems. For a proof-of-concept demonstration, we show in Fig. 6a, b the different ways we can write, adjust, erase and/or rewrite refractive index changes to create a designed phase

element. In Fig. 6b, the structures take the simple form of a 2D square phase object obtained by creating dot arrays with a 2-µm pitch corresponding to the modified voxel size in this case. A quantitative phase image of the object is then acquired using a custom infrared microscopy setup based on the phase-shifting method (details in Supplementary Note 8). With the images in Fig. 6b, we show how the phase can be gradually increased by using different writing energies, and decreased by using different nanosecond pulse energies so that a targeted phase level can be obtained inside Si. A quantitative analysis on these zones leads to the conclusion that phase levels exceeding 1 radian are accessible for one processed plane while maintaining a relatively good uniformity despite the discrete spot-to-spot irradiation method applied here. The uniformity is obviously an aspect which can be optimised with the laser scanning procedures and spot-to-spot overlaps. However, by comparing the first and second row of Fig. 6b, we found that the method of adjusting the phase by thermal erasure tends also to smooth the modified planes as the standard deviation is measured under 0.1 radians for all processed zone (values given in caption of Fig. 6b).

For a first functionality demonstration, we show in Fig. 6a a Fresnel lens written in a single layer inside a silicon wafer (depth of 300 µm), erased and re-written with a different design (spatial characteristics and index variations) so that it exhibits a different focal length as confirmed by the measurements. For a tunability demonstration, we create and adjust a step-phase-plate. As shown in Fig. 6c, the plate adds a phase delay to one half of an incoming focused beam to obtain a two-spot focus. The ratio between the two focal spots is then adjustable by changing the phase delay controlled by the writing parameters. For demonstration, we show in Fig. 6c measurements for a phase plate successively reconfigured to form two-spots with ratios up to more than 2:1. The results correspond well with the design expectations from propagation simulations (detailed in Supplementary Note 8). While we have concentrated here on reconfigurable flat-optics fabrication for demonstration purposes, such an approach can be beneficial for fine tuning or corrections in integrated photonics circuits fabricated by laser writing or other methods. Integrated quantum photonics surely offers a promising field of applications as reconfigurability is an essential feature to allow feedback and adaptive control, crucial for deterministic quantum teleportation, training of neural networks, and stabilization of complex circuits. In some of the most advanced solutions, reconfigurability is achieved by thermal[33] or electro-mechanical[39] stimulations. Here we demonstrate a laser-based monolithic technology adding to the panel of available solutions.

## Beyond monolithic silicon processing

We have concentrated here on solving the important controllability limitations of 3D writing in silicon, a fundamental material for electronics, and we have performed proof-of-concept experiments supporting a critical step towards silicon photonic applications like on-demand wavefront control and in-chip photonic memory devices by monolithic architectures. However, it is important to note that we did not only reach a writing performance similar to the one demonstrated in dielectrics and the current base materials for laser fabricated micro-optics, but we also surpassed it in some aspects thanks to the critical-plasma seed concept applicable inside semiconductors. Unfortunately, the approach cannot be directly implemented in dielectrics using conventional femtosecond lasers emitting in the visible or near-infrared domain (Titanium or Ytterbium based medium) because the required energy densities to create critical plasma conditions ($n \sim 10^{21} \, cm^{-3}$) are intrinsically above the melting and modification threshold of wide bandgap materials. However, we extrapolate that a similar concept remains applicable in dielectrics using a two-colour approach. Relying on the wavelength scaling of the critical plasma

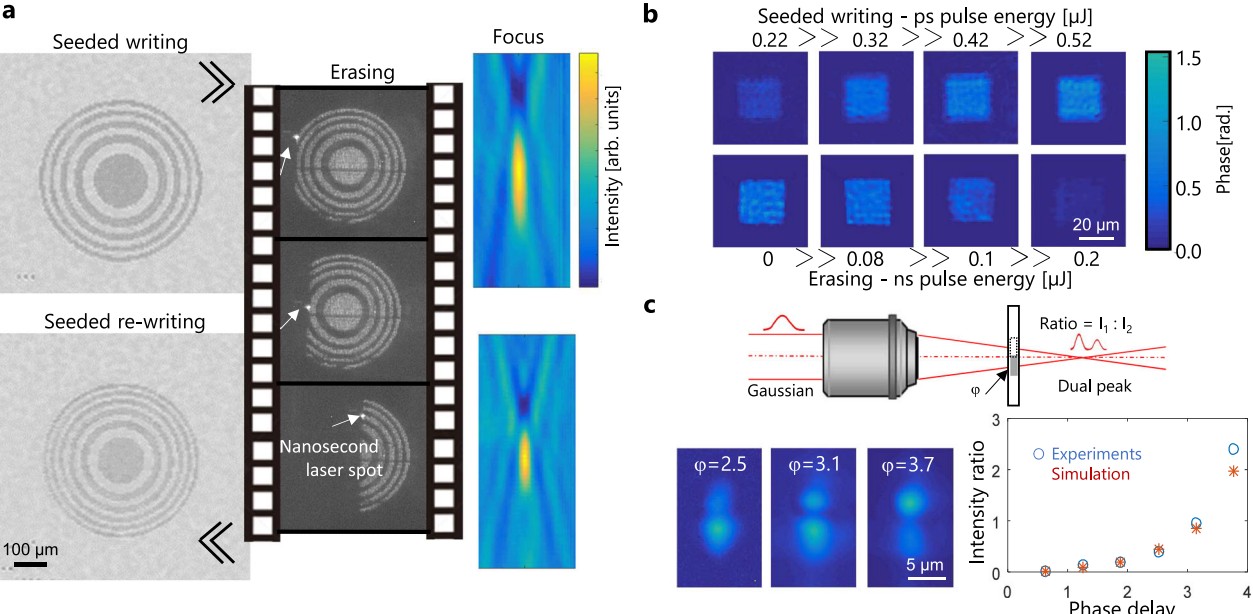

**Fig. 6 | Reconfigurable optical functions inside silicon by seeded writing.**
**a** Infrared microcopy images of a Fresnel plate written inside a silicon wafer (300-µm depth) and focusing performance characterization, followed by its erasure by scanned nanosecond laser irradiation (dark field images), and re-rewriting with design modifications for a change of the apparent focal length. **b** Infrared phase microscopy measurements of 2D phase plates at different phase levels obtained by seeded writing. The phase level can be varied by changing the picosecond pulse energy in the seeded writing procedure (first row) or by writing a given phase plate and progressively erasing it using nanosecond pulses of different energies (second row). The erasing leads to more uniform phase writing. The successive images of the second row reveal phase levels (and standard variation) measured at $0.81 \pm 0.02$, $0.74 \pm 0.09$, $0.61 \pm 0.07$, and $0.13 \pm 0.03$ radians. This shows the capacity to create phase elements which can be precisely reconfigured. **c** Measurements of the double-focus obtained by beam shaping with the step-phase plates produced by seeded writing. The ratio between the peak intensities of the spots compares well with those expected by simulation as a function of phase levels.

density, it remains realistic to create critical plasmas at reduced density levels provided that a mid-IR or THz radiation is introduced for the writing pulse. For instance, lasers emitting at wavelength above 5 µm are surely not optimal to deal with the diffraction-limit and thus are rarely considered for high resolution writing. However, they must permit interactions above the plasma wavelength of ionized regions well under $10^{20}$ cm$^{-3}$, imposing sub-diffraction limit longitudinal resolution (according to the Drude Model calculations, not shown). This must complement ideally the micrometre or submicrometric lateral resolution achievable for nonlinear pre-ionization with conventional femtosecond lasers. Taken all these considerations together, this leads to a very general concept for a new range of interaction experiments in any bandgap materials and a complementary role for the rapidly developing mid-infrared laser technologies to boost various process technologies.

## Methods
### Ultrafast laser interactions
An illustration of the setup is included in the Supplementary Note 1. The experiments are carried out with a femtosecond Ytterbium laser (Pharos, Light Conversion) combined with a high-power optical parametric amplifier (ORPHEUS-HP, Light Conversion) that delivers 1550-nm wavelength pulses of ~190 fs (FWHM) as measured using single-shot and scanning autocorrelators. In agreement with previous works, the generated ultrashort pulses with high temporal contrast are inoperative for permanent modification inside silicon due to strong nonlinear propagation effects delocalizing the radiation[10]. These include a strong plasma optical response that inspired the solution reported in this paper. A beam splitter is used to inject a portion of the beam energy in a stretcher arrangement with two Littrow gratings introducing negative group velocity dispersion to the original laser femtosecond pulses. The details of the stretcher configuration are discussed elsewhere[22]. For all experiments, the grating separation distance is set for the preparation of 11-ps pulses (FWHM) as measured by a long scan autocorrelator. Internal structuring of silicon is achievable with these stretched pulses tightly focused using a 0.85 NA infrared microscope objective (Olympus, LCPLN100XIR). For this pulse, the energy threshold for modification is ~0.33 µJ[22]. For tentatively tailoring energy delivery from the writing beam by plasma-based optical elements in near or intermediate fields, we use a motorized delay line to synchronize a pre-ionizing femtosecond pulse having the same polarization as the writing pulse. The configuration exploited for resolution and controllability enhancement uses a femtosecond pre-pulse of 150-nJ energy with a typical delay between pulses of 20 ps. This guaranties the interaction with highly contrasted plasma density distributions before typical nanosecond carrier diffusion processes come into play[40]. The pulses are combined using thin-film optics to create the double-pulse colinear beam directed toward the same focusing optics. For the interaction studies, the pulses are focused at a depth of 300 µm inside silicon samples and repeated irradiation is performed at low repetition rates (1 kHz) to avoid transient accumulation effects. The depth is chosen to ensure that the laser fluence on the surface is low (<GW cm$^{-2}$) for all tested situations and thus avoids surface effects influencing the bulk interactions. The objective is equipped with a correction collar, which is systematically adjusted for pre-compensating spherical aberration according to the changed thickness in all 3D writing demonstrations (see technical details on the writing procedures given in Supplementary Note 2). The theoretical linear point spread functions (PSF) for the expected non-aberrated beams (linear) are confirmed by delivered 3D fluence distribution measurements (details below) performed at low intensity.

### Interaction modelling
To simulate the interactions, we use an approach based on the method developed in ref. 23. For the effects with high-NA focusing conditions,

the simulations use a transformation optics approach detailed in ref. 7. The nonlinear propagation simulations are performed in two steps. First step describes the interaction of the pre-ionizing femtosecond pulse from which we extract the created plasma density distributions potentially influencing the writing beam. In the second step, we simulate the propagation and the added nonlinear ionization and energy transfer to the material for a picosecond writing beam. Because of a 3D simulation method leading to an exponential growth of computational resources with increasing pulse durations, we limit the pulse duration at 2 ps in the simulations. While this does not describe quantitatively the absolute energy density which is deposited, it allows to visualize the progressive hyper-localization of the energy flux. Laser and material parameters applied in the simulations and the results identifying the plasma-based features which are enhancing writing performances are described in Supplementary Note 4. Interestingly the simulations allow not only to calculate the final results of the experimental double-pulse irradiation but also the transient state created with the pre-ionization pulse. An Auger decay process consistent with time-resolved experimental measurements is then accounted to describe the free-carrier density evolution between pulses (see Supplementary Note 4). Systematic comparisons are made between the final delivered fluence distributions after simulated propagation of the writing beam with and without plasma assistance.

### Infrared imaging of microplasmas and modifications

We use a pump and probe lateral microscopy setup to measure by free-carrier absorption the characteristics of the micro-plasmas induced inside Si with our pre-pulses. The experimental arrangement based on an InGaAs array detector (Raptor, OWL SWIR 640.) is similar to that for the study of the fast kinetics of free-carriers injected in bulk Si by two-photon absorption[9]. For microplasma observations, the optical delay between the pump and probe pulses is set at only 10 ps so that we observe the free-carrier distributions before any significant decay[40]. For the detection and characterization of permanent modifications in the Si samples, the pump is blocked after the illumination. For phase microscopy of the modifications, we rely on the longitudinal differential interferometry technique[41] translated in the infrared domain of the spectrum. Experimental details are provided in Supplementary Note 8. For amplitude bright field imaging, we replace the ultrafast probe by a non-coherent uniform illumination (Quartz-Tungsten Halogen lamp) for improved bright-field imaging performance.

### Samples

For the microplasma and modification imaging experiments, we used microelectronics grade high-resistivity silicon crystals (orientation (100) < 0.5°, FZ, resistivity > 900 $\Omega$ cm to an initial free electron density <$5 \times 10^{12}$ cm$^{-3}$). Interactions are observed at a typical depth of 300-$\mu$m below the surface facing the pump beam. For high precision studies of the microplasma conditions, some imaging experiments are repeated using a solid-immersion focusing configuration (e.g., Fig. 1b). For this, we use 2-mm diameter spheres made of high resistivity float zone Silicon (resistivity > 10 k$\Omega$ cm, Tydex) and focus the light at the centre. Accordingly, by using a 0.3 NA microscope objective to focus the laser in the sphere, we can reproduce inside the sphere conditions similar to those obtained at NA = 0.85 inside the wafers. The working distance of the modest NA allows approaching the imaging objective for plasma observation. The sphere configuration also enhances the imaging performance leading to an advantage for high-resolution observations. Details on the technical aspects of this configuration can be found in a previous work (and its supplementary information)[11].

## Data availability

The data that support the results of this paper are available within the manuscript or supplementary information.

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

## Acknowledgements

This research has been conducted using LaMP facilities at LP3. The project has been supported by the European Union's Horizon 2020 research and innovation programme grant agreement No 724480 from the European Research Council (ERC), the French National Research Agency (ANR-22-CE92-0057-01) and the International Research Project (IRP) "MINOS".

## Author contributions

A.W., A.D., and P.S. performed the experiments. V.F and S.T. developed the numerical model and performed the theoretical study. D.G. led the design of the research. All authors interpreted the results and contributed to the manuscript originally prepared by A.W. and D.G.

## Competing interests
The authors declare no competing interests.
