## [Transparent Peer Review file · Nature Communications]

In-chip critical plasma seeds for laser writing of reconfigurable silicon photonics systems

Corresponding Author: Dr David Grojo

Version 1:

Reviewer comments:

Reviewer #1

(Remarks to the Author)

When writing in the bulk of semiconductors, challenges arise when tightly focusing fs laser pulses. The plasma rapidly saturates and no permanent modification is achieved.

So far, a method to circumvent this difficulty is to use longer pulses (ps) so that sufficient energy is deposited to reach material breakdown. However, this comes with a reduced resolution well beyond the diffraction limit as shown in this paper.

To overcome this challenge, the authors propose a method that consists in using femtosecond laser pulse as a seed, followed by a 20 ps pulse that effectively modifies the material. They noticed that such schemes lead to a much higher resolution and size-confinement of the modifications. The fs pulse creates a plasma that act as a shield for the incoming ps pulse, further confining the latter in a much smaller volume than would normally be expected. The coupling between the two pulses is shown by a convincing experiment where the authors shifts the focus point of the two beams, both laterely and vertically. The authors later on propose an analysis of the modified structures, using TEM and Raman, demonstrating some localized amorphization in the form of modified 'nano-volumes'. Finally, they report a mechanism of erasure by thermal annealing and exploit this property to demonstrate rewritable media, with a set of QR codes. They also demonstrate rewritable photonics components based on the same principle.

To our opinion, this is a significant addition to the ongoing effort towards efficient process for laser nanoscale manufacturing and an important contribution. The results are convincing and includes all aspects, from the concept, to its implementation, some modeling and finally, some illustrations of possible use of the technology. Hence, we feel that the paper is suited for publication in this journal after some revisions.

Below are specific comments on the paper.

Fig 1a is a bit confusing. Left suggests (fs pulse followed by ps pulse, which is correct) while the right side suggests the opposite.

Fig. 2b Which green line are we talking about? Confusing.

Fig. 2g. What is the time incremental difference between modifications?

Line 49 - 'Most demanding semiconductor applications'... A bit unexpected wording in a paper ('ad/sale speech writing style'). So, what are the 'most demanding semiconductor applications'?

Lines 137 - Add citations 'Previous works' (what previous works are we talking about?)

Line 202 - 'This already illustrates a clear potnetial for counterfeit marking applications.' Why this statement in this part of the text.

It seems a bit isolated and out of context and/or not clearly connected to the previous statemennts.

Lines 218-221 - The author could discuss similarities with the so called Type X reported by Prof. P. Kazansky's group (<https://doi.org/10.1038/s41377-020-0250-y>).

Lines 235-237 - Amorphization under femtosecond laser exposure is expected and has been reported in various systems (quartz, sapphire, etc.). Why would it be so surprising in silicon?

[see for instance: T. Gorelik, M. Will, S. Nolte, A. Tuennermann, and U. Glatzel, "Transmission electron microscopy studies

of femtosecond laser induced modifications in quartz," Applied Physics A: Materials Science & Processing 76, 309–311 (2003) // S. Juodkazis, K. Nishimura, H. Misawa, T. Ebisui, R. Waki, S. Matsuo, and T. Okada, "Control over the Crystalline State of Sapphire," Adv. Mater. 18, 1361–1364 (2006).]

Lines 255-256 - 'Due to the presence of dislocation'... Can we really talk about dislocations in this particular case?

General comments: while we see the value of storing information, for instance for QC controls and counterfeiting, it is unclear 'what rewritable photonics components' would practically entail and if it could be implemented in real applications. We do not quite understand the use case. Perhaps the authors can elaborate on this.

The author should also discuss how generic is the method. Could we transpose it to other material systems?

In general, the quality of the figures may have suffer pdf conversions. We would recommend double-checking the original quality.

Reviewer #2

(Remarks to the Author)

Silicon is one of the most important materials for advanced photonic and electronic devices. There have been numerous attempts for 3D inscription within silicon to improve the integration density and structural simplicity. However, unlike glass, precise control of internal modification of diffraction limited features has been difficult due to the mixed linear and nonlinear optical phenomena when using NIR ultrafast lasers.

This manuscript provides a nice way to utilize carefully paired double ultrashort pulses that can induce much smaller damage inside silicon than permanent damage that is achieved by a single picosecond pulse. This modification is possible by the assistance of first pulse to achieve the critical plasma density near focal region in silicon prior to the second pulse reaching its focus where the critical plasma density by the first pulse, preventing the second pulse from further propagation. The authors not only showed that the damage inside silicon modified in this way can be reversibly erased and rewritten, but also further demonstrated that this modification method can inscribe structured information by inducing controllable highly localized damage arrays inside silicon.

The reviewer believes that this work will have impact on opening a new door for 3D silicon technology to overcome the limitation of 2D-based technologies. Therefore, I recommend this manuscript to be published in this prestigious journal. However, before publication, I would like the authors to improve the manuscript by clarifying a few ambiguities to help readers understand more precisely.

1. The authors claim that the two-pulse modification within a silicon crystal is reconfigurable by erasing with a ns laser. However, it is still not clear how the modified site where permanent damage or amorphization by laser pulses exist can be healed (or perfectly re-crystallized). Is it the "local annealing or remelting" process followed by "relatively slow quenching" to recrystallize?
2. Can any evidence of TEM or Raman analysis of the erased site be provided after erasing? The erasing potential is rapidly decreased after the pulse number of 80~90(S7-7). I am curious if there is certain defect density that becomes a trigger to prevent further erasing process.
3. How is the interval between fs and ps pulse determined? What is the most dominant factor in determining τ ?
4. I strongly suggest to improve the overall quality of figures and their captions, and if needed, separate some figures to improve their visibility.

Reviewer #3

(Remarks to the Author)

The paper by Wang et al. explores controlled 3D laser micro-fabrication buried inside silicon (i.e., in-chip/in-wafer technology). The novelty claim highlights the first achievement of sub-diffraction-limit fabrication resolution inside Si created with ultrafast laser writing.

While the paper is interesting for technical reasons, both the 3D micro-scale fabrication and controlled nano-fabrication has been demonstrated inside Si with lasers. Indeed, it is not clear which fabrication metric the paper is compared with the state-of-the-art from novelty perspective. For instance, resolution of 100 nm has been shown (Sabet et al., Nature Commun., 2024). Similarly, various double-pulse/beam writing techniques inside silicon are given (Chambonneau et al., Laser Photon. Rev. 15, 2100140, 2021).

Further comments on the strength and weaknesses of the manuscript are provided below:

Strengths:

- 1- The paper achieves 3D micro-controlled subsurface fabrication in Si. While the reported fabrication resolution of 1.6 μm modestly improves upon compared to the picosecond case of 6 μm , the 3D micro-scale control capability may have practical value.

2- The concept of critical plasma as driver is interesting and can potentially find applications in micro-fabrication.

3- The observation of subsurface amorphous regions upon irradiation confirms previous similar observations. However in the current form, their size, location and/or periodicity is not controlled.

Weaknesses:

1- The paper reports a lateral resolution of 1.6 μm and longitudinal resolution of 2.6 μm inside Si. Considering that the wavelength of laser in Si is $\sim 450\text{ nm}$ ($1500\text{ nm}/3.5$), the reported performance is significantly lower than allowed by the wavelength. Further, if one considers the diffraction limit as $\lambda/2 = 225\text{ nm}$, the main claim of the paper, i.e., achieving sub-diffraction fabrication resolution seems incorrect. Indeed, the written structures have feature sizes that are an order of magnitude above the diffraction limit.

2- The novelty claims requires an overhaul, removing the incorrect sub-diffraction claims and perhaps focusing on the ultrafast 3D micro-fabrication capability and the physics of the plasma seeding.

3- In relation to the preceding considerations, the state-of-the-art of nano-fabrication inside silicon is not discussed in the paper (Sabet et al, Nature Commun., 15, 5786, 2024).

4- Line 58 suggests "super-resolution writing" inside silicon and Line 113 claims to prove this super-resolution fabrication. However, similar micro-scale arrays have already been shown multiple times in the literature (e.g, Figs. 31, 37, 41, Chambonneau, Laser Photon Rev 15, 2100140, 2021). It is not clear why such above diffraction-limit structures are repeatedly identified as "super-resolution".

5- The authors do not provide feature size analysis from any highly periodic patterns, or fabricate periodic continuous lines or planes, but repeatedly claim "superior precision". What is the minimum feature size of such line/plane patterns, the minimum period between any two lines, planes or voxels? Further, there is no standard deviation analysis, error bars and roughness analysis. Such measurements are critical for performance evaluation of any 3D fabrication method.

6- The 3D Eiffel Tower micro-pattern could also be fabricated with using only ps pulses - as indeed confirmed in the supplementary material (both ps and fs/ps cases would still be above the diffraction limit, with the former requiring somewhat larger volume, pointing to a moderate advance).

7- Dual-pulse ultrafast experiments using parallel and orthogonal polarization is reported (Shimotsuma, Y. et al. Appl. Phys. A 122, 159, 2016). The reported experimental concept using dual pulses seem to be a similar technique.

8- The statement of Line 125 is not convincing, claiming a "realization requiring a level of precision and flexibility out-of-reach for the most advanced single beam writing configurations". In particular inside glass, there are numerous examples at similar precision and flexibility (e.g., Lei et al, Optica, 8, 11, 1365, 2021; Lancry et al, Laser Photon Rev. 7, 953, 2013; Delgoffe et al, Optica, 10, 10, 2023).

9- The claim of achieving "the first clear evidence that amorphization in the bulk silicon" is confusing. Laser amorphization inside Si is reported in a number of articles and may be cited: Wang et al. J. Laser Appl. 36, 022015, 2024; Verburg et al., Appl. Phys A., 120, 683, 2015).

Other comments:

1- What are the beam propagation directions in Fig. 2a, is this not critical for the model? The description in the caption of Fig. 2d does not match to the conceptual illustration in Fig. 2d. There is a phrase "pointing" in the figure, do the authors mean Poynting vector? Which color corresponds to which laser here? The associated plasma physics and its description is not clear in the text.

2- Line 180 seems incorrect, $t_{fs}-t_{ps}$ = negative values would correspond to fs preionization. The discussion in Lines 185-187 (about pulse splitting and its interactions) is not clear.

3- Are there any subsurface modifications that are not visible to the microscope images between the voxels effecting size measurements?

4- Line 189: "Unprecedented high-degree of ionization" claim is given, but not clear with respect to what? The manuscript may benefit by refraining from ambiguous/unclear claims.

5- The following claim on reconfigurability may refer to previous work: "a new type of structures (see above) that permit near-perfect erasure by thermal annealing" Similar structures and thermal erasing is mentioned in page 21 of Chambonneau, Laser Photon Rev 15, 2100140, 2021.

6- The anisotropy observed in Fig. 3d-e may be due to other effects such as birefringence. A more direct evidence would be

required.

7- The paper mentions spherical aberration compensation to maintain uniformity for writing at different depths. However, the discussion of depth control is quite limited.

Version 2:

Reviewer comments:

Reviewer #1

(Remarks to the Author)

I am satisfied with the answers provided by the authors.

The work offers some novel contributions to the existing body of published papers on the topic. It is a broad interest for a large audience and will certainly stimulate further research on the topic, in particular on how to expand these methods towards other substrates.

Reviewer #3

(Remarks to the Author)

The revised manuscript represents a meaningful technical advance in the field of ultrafast laser–material interactions, with direct relevance to semiconductor photonics. Following the revision, the central message of the paper is more clearly articulated and better supported by experimental evidence. I outline one major point that warrants further attention, along with several additional comments below.

While the current manuscript offers a well-executed and experimentally rich contribution, it builds on earlier work by the authors (Phys. Rev. Res. 2, 033023, 2020), which had introduced the synergistic use of femtosecond and picosecond pulses to reduce modification thresholds in silicon. That study presents the foundational idea that an ultrafast pulse arriving shortly before another can prime the medium by injecting free carriers, enabling localized energy deposition. The present work extends this approach by intentionally engineering the plasma seeding conditions and demonstrates their utility in functional structuring. It would strengthen the manuscript if the authors could further elaborate on how the scope and mechanism of the present study build upon and differ from the earlier publication.

The authors also demonstrate a range of functional outcomes: rewritable 3D optics, data storage, and tunable phase plates in silicon. The combination of rewriting capability and high repeatability represents a valuable advance. The integration of simulations, pump-probe microscopy, and application-oriented demonstrations provides support for the demonstrated approach.

Other comments:

- The QR codes shown in the manuscript do not appear to open or link successfully, returning a 404 error. This may be due to embedded images or limited resolution in the PDF version.
- On page 15, the statement that “any phase value is achievable” is vague and potentially overstated—e.g., a full 2π phase shift is not explicitly demonstrated. Similarly, for the statement on page 17 regarding “on-chip photonic memory devices never envisioned before”.
- Are the 2-mm spherical-shaped samples used for a particular reason.

Version 3:

Reviewer comments:

Reviewer #3

(Remarks to the Author)

The Reviewer is satisfied with the edits and thanks the authors.

RESPONSE TO REVIEWER COMMENTS

Reviewer #1 (Remarks to the Author):

When writing in the bulk of semiconductors, challenges arise when tightly focusing fs laser pulses. The plasma rapidly saturates and no permanent modification is achieved.

So far, a method to circumvent this difficulty is to use longer pulses (ps) so that sufficient energy is deposited to reach material breakdown. However, this comes with a reduced resolution well beyond the diffraction limit as shown in this paper.

To overcome this challenge, the authors propose a method that consists in using femtosecond laser pulse as a seed, followed by a 20 ps pulse that effectively modifies the material. They noticed that such schemes lead to a much higher resolution and size-confinement of the modifications. The fs pulse creates a plasma that act as a shield for the incoming ps pulse, further confining the latter in a much smaller volume than would normally be expected. The coupling between the two pulses is shown by a convincing experiment where the authors shifts the focus point of the two beams, both laterely and vertically. The authors later on propose an analysis of the modified structures, using TEM and Raman, demonstrating some localized amorphization in the form of modified 'nano-volumes'. Finally, they report a mechanism of erasure by thermal annealing and exploit this property to demonstrate rewritable media, with a set of QR codes. They also demonstrate rewritable photonics components based on the same principle.

To our opinion, this is a significant addition to the ongoing effort towards efficient process for laser nanoscale manufacturing and an important contribution. The results are convincing and includes all aspects, from the concept, to its implementation, some modeling and finally, some illustrations of possible use of the technology. Hence, we feel that the paper is suited for publication in this journal after some revisions.

We are pleased by the very positive opinion on the interest and quality of our work. We thank the reviewer for her/his time and the following comments that allow us to even further improve the paper. We provide below our point-by-point responses.

Below are specific comments on the paper.

Fig 1a is a bit confusing. Left suggests (fs pulse followed by ps pulse, which is correct) while the right side suggests the opposite.

We acknowledge some figures in the original manuscript were confusing and may lack quality. In the revised version, we have largely modified the figure presentations (same content now splitted in 6 figures) to consider this point and other requests. We paid particular attention to properly illustrate our configuration, which we confirm is: a fs pulse followed by a ps pulse.

Fig. 2b Which green line are we talking about? Confusing.

We apologize for this other mistake on the figures. The line positioned at the plasma level determined by the measurements and used to extrapolate the expected penetration depth was simply missing in the submitted version of the figure. In the revisions for improved figure

presentation, we have actually suppressed this figure. The prediction using a simple Drude model is now directly given in the text without referring to a figure (see page 7 of marked version).

Fig. 2g. What is the time incremental difference between modifications?

The different modifications are systematically produced and presented according to a constant time increment. With the original figure showing the “zero” and only the most extreme delay values, we acknowledge a lack of clarity on this aspect. In the new version (new fig 3e), we add ticks and labels to clearly show the applied linear scales with 40-ps increments between modifications for the long-delay scan (top image) and 2-ps increments for the short-delay scan (bottom image).

Line 49 - 'Most demanding semiconductor applications'... A bit unexpected wording in a paper ('ad/sale speech writing style'). So, what are the 'most demanding semiconductor applications'?

We acknowledge an inappropriate expression and simply replace ‘... impractical for the most demanding applications’ by ‘hardly applicable for real device fabrication’. The reworked previous sentences state more explicitly the challenges to improve the “controllability” and “resolution” needed to be relevant for semiconductor technologies (see marked version of the manuscript, page 2).

Lines 137 - Add citations 'Previous works' (what previous works are we talking about?)

For the pioneer works evidencing of low-density plasmas in a large region in the pre-focal zone, and subsequent changes in beam propagation, we simply add the references 16 and 17 (now 8 and 9) of the original manuscript. Since these works, numerous other direct or indirect evidences have been discussed. Thus, we cite also at this location the review paper given as reference 6 (now 10) to fully cover the literature on this question (see marked version of the manuscript, page 6).

8. Kononenko, V. V, Konov, V. V & Dianov, E. M. Delocalization of femtosecond radiation in silicon. *Opt Lett* 37, 3369–71 (2012).

9. Mouskeftaras, A. et al. Self-limited underdense microplasmas in bulk silicon induced by ultrashort laser pulses. *Appl Phys Lett* 105, 191103 (2014).

10. Chambonneau, M. et al. In-Volume Laser Direct Writing of Silicon—Challenges and Opportunities. *Laser Photon Rev* 15, 2100140 (2021).

Line 202 - 'This already illustrates a clear potential for counterfeit marking applications.' Why this statement in this part of the text. It seems a bit isolated and out of context and/or not clearly connected to the previous statements.

We agree that it is a too focused application at this stage of the manuscript. This sentence is simply suppressed as the potential for “counterfeiting or traceability measures in the semiconductor industry” is already mentioned in the reconfigurability part.

Lines 218-221 - The author could discuss similarities with the so called Type X reported by Prof. P. Kazansky's group (<https://doi.org/10.1038/s41377-020-0250-y>).

We acknowledge that making a parallel with other low-loss structures demonstrated in dielectrics is interesting and can shed light on the reported observations in semiconductors. We add a comment

supported by the suggested reference in the text of the revised manuscript (see marked version of the manuscript, page 11 and 12). In particular, we emphasize that birefringence is expected with the demonstrated anisotropy but it was too low to be measurable in silicon.

Lines 235-237 - Amorphization under femtosecond laser exposure is expected and has been reported in various systems (quartz, sapphire, etc.). Why would it be so surprising in silicon?

[see for instance: T. Gorelik, M. Will, S. Nolte, A. Tuennermann, and U. Glatzel, "Transmission electron microscopy studies of femtosecond laser induced modifications in quartz," *Applied Physics A: Materials Science & Processing* 76, 309–311 (2003) // S. Juodkazis, K. Nishimura, H. Misawa, T. Ebisui, R. Waki, S. Matsuo, and T. Okada, "Control over the Crystalline State of Sapphire," *Adv. Mater.* 18, 1361–1364 (2006).]

In view of the literature in dielectrics (incl. the papers cited here), we acknowledge that this specificity of silicon may not appear completely intuitive depending on the readers. However, we confirm that the writing of fully-amorphous spots in bulk silicon would be highly desirable but has not been achieved to date. By femtosecond laser interaction on silicon surfaces, the ultrafast quenching conditions lead to the formation of well-defined amorphous layers. However, all attempts to translate similar conditions in the bulk and to achieve the same transformation with 3D control have simply failed to date. Various types of internal modifications in different regimes have been the subject of TEM and electron diffraction analyses. While a partial amorphization was searched in most cases to explain the apparent positive refractive index change found with the modifications, all studies (including ours) observed mainly stressed monocrystalline material. Indirect evidence of the amorphous material is discussed by extrapolation from Raman spectra and rare amorphous grains have been reported recently. However, this supports only a residual amorphization with <0.2% transformed into this state [Trinh, L., Wang, X., Zhang, X., Hosseini-zavareh, S. & Mao, A. Transmission Electron Microscopy Characterizations of Local Amorphization of Single Crystal Silicon by Nanosecond Pulsed Laser Direct Writing. 1–7 (2023) doi:10.1002/adem.202301377.]. In our work, we do not transform the entire voxel into an amorphous state but we report for the first time on a significant amount of amorphized silicon in the processed zone. We believe this corresponds to an important advance associated with different conditions achieved in the material with our irradiation scheme. We thank the reviewer for pointing out this important question that we discuss in more detail with references in the revised manuscript (see page 12 of the marked version).

Lines 255-256 - 'Due to the presence of dislocation'... Can we really talk about dislocations in this particular case?

We acknowledge the wording "dislocation" designates a specific crystallographic feature that may appear confusing. In this case, we describe ruptures and discontinuities in the microscopic responses. Then, we replace this term by "material disruption" in the revised version of the manuscript (see marked version of the manuscript, page 11 and 13).

General comments: while we see the value of storing information, for instance for QC controls and counterfeiting, it is unclear 'what rewritable photonics components' would practically entail and if it could be implemented in real applications. We do not quite understand the use case. Perhaps the authors can elaborate on this.

Reconfigurable photonics is the basis for numerous exciting opportunities in applications such as actively controlled light modulation, optical power limiting and rerouting, tunable displays, active spectral filtering, and dynamic wavefront shaping. This is today an extensive field of research primary

concentrating on material science aspects. The exploitation of thin layers of so-called phase-change materials (PCM) for reversible refractive engineering is the basis of many advanced demonstrations [a,b]. In this context, we believe it is particularly interesting to reveal the possible direct use of crystalline silicon to achieve similar technologies with appropriate irradiations.

A second rationale can be also introduced on the basis of the manufacturing process so called “Laser trimming” which consists in using a laser to adjust the operating parameters of an electronic circuit. The way we can readjust the refractive index and/or locally erase with three-dimensional control in a chip gives also the vision of a similar technology applicable to Silicon Photonics. While our paper concentrates on reconfigurable flat-optics fabrication for demonstration purposes, such an approach can be beneficial for fine tuning or corrections in integrated photonics circuits fabricated by laser writing or other methods. Integrated quantum photonics offers a promising field of applications as reconfigurability is an essential feature to allow feedback and adaptive control, crucial for deterministic quantum teleportation, training of neural networks, and stabilization of complex circuits. In some of the most advanced solutions, reconfigurability is achieved by thermal [c] or electro-mechanical [d] stimulations. Here we demonstrate reconfigurability by direct laser writing of silicon for a monolithic technology adding to the panel of available solutions.

In page 12 of the manuscript (see marked version), we add the comments on the potential perspectives of application for integrated photonics with the corresponding references [c,d] (new references 39 and 40).

[a] Cueff, S. *et al.* Reconfigurable Flat Optics with Programmable Reflection Amplitude Using Lithography-Free Phase-Change Material Ultra-Thin Films. *Adv. Opt. Mater.* **9**, 1–11 (2021).

[b] Menshikov, E. *et al.* Reversible Laser Imprinting of Phase Change Photonic Structures in Integrated Waveguides. (2024) doi:10.1021/acsami.4c04573.

[c,39] Ambrosio, V. D. *et al.* Thermally reconfigurable quantum photonic circuits at telecom wavelength by femtosecond laser micromachining. *Light Sci. Appl.* 1–7 (2015) doi:10.1038/lisa.2015.127.

[d,40] Gyger, S. *et al.* Reconfigurable photonics with on-chip single-photon detectors. *Nat. Commun.* **12**, 1–8 (2021).

The author should also discuss how generic is the method. Could we transpose it to other material systems?

While we believe the method must be applicable to some other materials, it is difficult from our work concentrating on silicon to emphasize on a potential universality. Given the importance of this comment for broad interest and impact, we have discussed in the original manuscript the question of conditions for enhanced spatial resolution with the proposed concept of critical plasma seeds depending on materials. In particular, we have extrapolated the method must be applicable to wide band gap dielectrics provided Mid-IR lasers are introduced, as well as other semiconductors having similar properties like silicon (i.e. GaAs, InP). To highlight the discussions, we have added a title for the last section “Beyond monolithic silicon processing” and have made small adjustments in the text (see page 17 of marked manuscript).

Regarding the reversibility and tuning of the written structures, we believe it would be much more difficult to discuss the generality as we rely in the present case on defect accumulations which can be thermally relaxed in silicon. While future works may concentrate on these material science questions leading to amorphous grain formations, we believe it is the understanding of the nature of initially induced single defects/color centers in silicon which is crucial to evaluate the applicability in some

other materials. To avoid unnecessary speculations on these interesting but also complex questions, we prefer not to comment on this aspect in the main text of the manuscript.

In general, the quality of the figures may have suffer pdf conversions. We would recommend double-checking the original quality.

As commented before, we acknowledge a general lack of quality in the figures. We have reworked the figure presentations and paid more attention to the quality after pdf conversions.

Reviewer #2 (Remarks to the Author):

Silicon is one of the most important materials for advanced photonic and electronic devices. There have been numerous attempts for 3D inscription within silicon to improve the integration density and structural simplicity. However, unlike glass, precise control of internal modification of diffraction limited features has been difficult due to the mixed linear and nonlinear optical phenomena when using NIR ultrafast lasers.

This manuscript provides a nice way to utilize carefully paired double ultrashort pulses that can induce much smaller damage inside silicon than permanent damage that is achieved by a single picosecond pulse. This modification is possible by the assistance of first pulse to achieve the critical plasma density near focal region in silicon prior to the second pulse reaching its focus where the critical plasma density by the first pulse, preventing the second pulse from further propagation. The authors not only showed that the damage inside silicon modified in this way can be reversibly erased and rewritten, but also further demonstrated that this modification method can inscribe structured information by inducing controllable highly localized damage arrays inside silicon. The reviewer believes that this work will have impact on opening a new door for 3D silicon technology to overcome the limitation of 2D-based technologies. Therefore, I recommend this manuscript to be published in this prestigious journal.

We are very pleased by these positive comments. We thank the reviewer for the requested clarifications that allow us to even better convey our findings to the reader. We provide below point-by-point responses indicating the applied changes.

However, before publication, I would like the authors to improve the manuscript by clarifying a few ambiguities to help readers understand more precisely.

1. The authors claim that the two-pulse modification within a silicon crystal is reconfigurable by erasing with a ns laser. However, it is still not clear how the modified site where permanent damage or amorphization by laser pulses exist can be healed (or perfectly re-crystallized). Is it the “local annealing or remelting” process followed by “relatively slow quenching” to recrystallize?

After re-reading our manuscript with this comment in mind we acknowledge a lack of clarity on this aspect. First, it is important to note that the amorphous nano-domains found in the TEM analyses correspond to irradiations with a large number of shots (>100). According to our other attempts for TEM analyses on the density when writing with a low number of pulses, it is very likely we rely on local structural defects without the presence of amorphous grains. The written structures which are efficiently erased by the nanosecond laser are also as easily erasable at ~1000°C for 4h in furnace so we believe it is not re-melting and recrystallization but we rely most likely on the annihilation of defect centers by thermal treatment in these demonstrations. To support this aspect, we can refer to ref (Quard et al. Phys. Rev. Appl., 21(4), 044014 (2024)) in which the selective erasure of G-centers produced by femtosecond laser irradiation of surfaces is observed by flash thermal treatment. In page 12 of the new version (see marked version), we now comment on this aspect.

2. Can any evidence of TEM or Raman analysis of the erased site be provided after erasing? The erasing potential is rapidly decreased after the pulse number of 80~90(S7-7). I am curious if there is certain defect density that becomes a trigger to prevent further erasing process.

We totally agree that the limitation for the cycling experiments (repeated writing and erasure) must rely on defect accumulation. Unfortunately cannot confirm by TEM or Raman analyses on these

optically “invisible” micrometer modified voxels. Because of the technical complexity for sample preparation (cross-sectioning of the modifications), the TEM and Raman analyses presented in this work have been only possible for silicon modified on a typical volume of $100\mu\text{m}^3$ by writing with a scanning procedure. These correspond to >100 applied pulses leading to a well contrasted modification which is not directly relevant for the conditions used for the cycling experiments. The latter rely on nearly “invisible” spots revealed by their scattering response only (not visible by transmission imaging). On the basis of all our attempts to characterize modest modifications, we are only able to conclude on the inability to reveal any characteristic defect either due to insufficient sensitivity of the applied characterization methods and/or extremely low density of created defects. In page 12 of the new version (see marked version), we now comment on these aspects.

3. How is the interval between fs and ps pulse determined? What is the most dominant factor in determining τ ?

As detailed in the manuscript, a crucial aspect to access the improved writing is to see the synchronized writing pulse interacting with a plasma front at near critical density. From pump-probe measurements and calculations, we have estimated the plasma density at $\sim 3 \times 10^{20} \text{ cm}^{-3}$ for the laser energy conditions of our experiments and the 20 ps delay. We have chosen this short delay to avoid significant plasma decay. As shown with Fig S4-1, we have measured a relatively fast decay of the plasma density likely due to an Auger process but it remains significantly longer than the chosen delay ($\sim 250\text{ps}$ at $1/e$). In our experiments, 20 ps is chosen for a short delay while avoiding temporal overlap with our pulses (10ps duration for the writing pulse). We confirm that similar results remain achievable in the range 10-50ps but the performance rapidly degrades for longer delays. For longer delays, there is no efficient possibility to compensate by increasing the femtosecond pre-pulse because of excitation conditions already at the saturation level for the studied conditions. Increasing femtosecond pre-pulse energy moves the plasma front in the backward direction more than enhancing its density as it is shown with new Fig 3c.

4. I strongly suggest to improve the overall quality of figures and their captions, and if needed, separate some figures to improve their visibility.

We acknowledge a lack of quality. To address this and other comments from the different reviewers, we have reworked the figure presentation with more attention paid to the caption and the quality after pdf conversions. As suggested, some figures have been split. The revised manuscript now includes 6 figures instead of 4 with the originally submitted version.

Reviewer #3 (Remarks to the Author):

The paper by Wang et al. explores controlled 3D laser micro-fabrication buried inside silicon (i.e., in-chip/in-wafer technology). The novelty claim highlights the first achievement of sub-diffraction-limit fabrication resolution inside Si created with ultrafast laser writing.

While the paper is interesting for technical reasons, both the 3D micro-scale fabrication and controlled nano-fabrication has been demonstrated inside Si with lasers. Indeed, it is not clear which fabrication metric the paper is compared with the state-of-the-art from novelty perspective. For instance, resolution of 100 nm has been shown (Sabet et al., Nature Commun., 2024). Similarly, various double-pulse/beam writing techniques inside silicon are given (Chambonneau et al., Laser Photon. Rev. 15, 2100140, 2021).

We would like to thank the reviewer for her/his time and constructive report. We are pleased by the positive comments on the technical values and interest of our work, but we also understand certain reservations in particular on the question of process resolution.

First, we would like to confess a mistake with the absence of reference to the important and highly relevant recent paper by Sabet et al. This is due to a paper published (16 July 2024) after our original submission (25 June 2024). Later, our paper was transferred to Nature Communications. We realize that we have actually commented this new work in our cover letter to the editor to explain the positioning and importance of our findings. However, we missed to incorporate these up-to-date considerations in the introduction of our transferred paper.

Briefly, our letter to the editor stated that the quest for a new laser technology for semiconductor applications leads us today to the recent publication from Tokel's group (Sabet et al., Nature Commun., 2024). This work impressively achieves nanometer precision but with the drawback of a need for pre-modified structures to reach this level of precision. In view of this recent work, our work surely does not immediately demonstrate improved resolution but it compares well given our digital direct-writing approach. We also explained in the letter that, more importantly, we complement ideally these important advances by the capacity to efficiently erase and reconfigure laser-induced modifications. This must open perspectives for reconfigurable 3D semiconductor devices by direct laser writing.

As can be seen in the revised version, this vision and the corresponding reference are now added in the introduction part (see marked version of the manuscript, page 2). In the following responses to the listed weaknesses, we also explain and detail the applied changes to better assess the achieved level of resolution.

Further comments on the strength and weaknesses of the manuscript are provided below:

Strengths:

S1- The paper achieves 3D micro-controlled subsurface fabrication in Si. While the reported fabrication resolution of 1.6 μm modestly improves upon compared to the picosecond case of 6 μm , the 3D micro-scale control capability may have practical value.

S2- The concept of critical plasma as driver is interesting and can potentially find applications in micro-fabrication.

S3- The observation of subsurface amorphous regions upon irradiation confirms previous similar observations. However in the current form, their size, location and/or periodicity is not controlled.

We thank the reviewer for this vision on the strengths of our report. Our responses given below also comment on the different aspects mentioned here.

Weaknesses:

W1- The paper reports a lateral resolution of 1.6 μm and longitudinal resolution of 2.6 μm inside Si. Considering that the wavelength of laser in Si is $\sim 450\text{ nm}$ ($1500\text{ nm}/3.5$), the reported performance is significantly lower than allowed by the wavelength. Further, if one considers the diffraction limit as $\lambda/2 = 225\text{ nm}$, the main claim of the paper, i.e., achieving sub-diffraction fabrication resolution seems incorrect. Indeed, the written structures have feature sizes that are an order of magnitude above the diffraction limit.

W2- The novelty claims requires an overhaul, removing the incorrect sub-diffraction claims and perhaps focusing on the ultrafast 3D micro-fabrication capability and the physics of the plasma seeding.

W3- In relation to the preceding considerations, the state-of-the-art of nano-fabrication inside silicon is not discussed in the paper (Sabet et al, Nature Commun., 15, 5786, 2024).

W4- Line 58 suggests "super-resolution writing" inside silicon and Line 113 claims to prove this super-resolution fabrication. However, similar micro-scale arrays have already been shown multiple times in the literature (e.g, Figs. 31, 37, 41, Chambonneau, Laser Photon Rev 15, 2100140, 2021). It is not clear why such above diffraction-limit structures are repeatedly identified as "super-resolution".

We treat together W1-4 all about the question of writing resolution.

As explained hereafter, we acknowledge that the resolution increase may first appear relatively modest and not necessarily corresponding to some terminologies as "super-resolution", "sub-diffraction" or "diffraction limited". Then, to avoid confusions, we have revised our manuscript to suppress all these terminologies and introduced the reference to the paper (Sabet et al., Nature Commun., 2024) which is surely more appropriate to support nanoscale writing capacities (see also previous comment).

However, we would like to comment on the significance of the achieved level of improvement. To assess the real level of resolution and the optical limits, we respectfully consider the diffraction limit given by the reviewer at 225nm is not immediately relevant. For a practical case, it is important to also consider the numerical aperture to define a limit. Then, unless one turns to solid-immersion focusing, there is no expected decrease of the spot size considering the refractive index of the material. This is because, by definition, the refraction at an air-interface (flat) exactly compensate the index change ($NA = \sin(\theta_{\text{air}}) = n_{\text{Si}}(\theta_{\text{Si}})$). In brief, when focusing a beam inside a material (flat interface), in comparison to vacuum, the lateral spot size remains unchanged, and it is only the confocal parameter which varies (multiplied by n). Experimental evidences of all these considerations for similar configurations can be found in refs. 5 and 7 of our original manuscript (refs. 7 and 11 in the revised version).

For the conditions of our experiments, Abbe's limit $\lambda/2NA$ is $\sim 1\mu\text{m}$. This can somehow describe the achievable spot size but also a limitation for the optical observation. Even if we observe modifications progressively vanishing in optical images, we then reasonably agree that it is preferable

to avoid any claim of a diffraction-limited precision. The discussion on the longitudinal resolution may be more reasonable as a 1- μm spot diameter translates in a confocal parameter $n \cdot 2\pi \cdot (D/2)^2 / \lambda$ of about $>3.5\mu\text{m}$, which is clearly above the measured features at $2.6\mu\text{m}$. However, we agree these are unnecessary considerations. In the text we now more simply express the demonstrated gains by using the critical plasma seeds: $>35\%$ in lateral resolution and $>75\%$ longitudinal resolution. These are significant numbers to convey the value of the method in comparison to previous works without the need for discussions on the optical limits.

In line with these revisions, we also suppress the optical limits on the graphs presented in original figure 1f (now 2b). Instead, we simply use the wavelength as a reference line to guide the eyes on the achievable level of performance with critical plasma seeds.

W5- The authors do not provide feature size analysis from any highly periodic patterns, or fabricate periodic continuous lines or planes, but repeatedly claim “superior precision”. What is the minimum feature size of such line/plane patterns, the minimum period between any two lines, planes or voxels? Further, there is no standard deviation analysis, error bars and roughness analysis. Such measurements are critical for performance evaluation of any 3D fabrication method.

W6- The 3D Eiffel Tower micro-pattern could also be fabricated with using only ps pulses - as indeed confirmed in the supplementary material (both ps and fs/ps cases would still be above the diffraction limit, with the former requiring somewhat larger volume, pointing to a moderate advance).

We treat together W5-6 regarding the corresponding performance evaluation and limits for applications.

We believe we can clarify most of these concerns with the realizations presented in our paper and supplementary information.

First, re-reading the manuscript with these comments in mind, we realized that several technical details, important for performance evaluation, were missing in the technical descriptions of supplementary Note 2 and the works presented in Fig. 2, 5 and 6 (including some laser conditions depending on cases). This has been corrected.

Then, we would like to emphasize the obvious differences in quality for the two realizations shown in Fig. S2-2c. Both have been performed with the best possible irradiation parameters. For critical plasma seed writing, the separation between each written voxel is $2\mu\text{m}$ that allows according to the spatial characteristics of the individual spots described above to merge modifications and so tentatively achieve uniform writing. For pure picosecond laser writing, despite energy conditions as close as possible to threshold conditions, we immediately see that the writing resolution limitations and the type of modifications (strongly opaque) cause important problems (discussed in section S2.2. and fig. S2-2e). Relying on gradual modifications with the critical plasma seeds, we unambiguously demonstrate a controllability superiority by these two images. Given the size of the realizations, we believe it is enough to claim a controllability at true micrometer level while the traditional picosecond writing is somehow limited at typical $\sim 10\mu\text{m}$ level for similar quality for such structure complexity.

We have not tried to repeat writing experiments of the same structure at smaller scale because of the spot characteristics described above and the difficulty of characterizations for less contrasted writing regimes as those discussed with the writing/erasing experiments. However, we can confirm that the pure-picosecond writing case has been the subject of various optimizations (energy, number

of applied pulses, spherical aberration compensations) leading to the conclusion of an incapacity to present a structure of comparable quality at this scale level.

Regarding the technical aspects, we have added the missing information about the writing procedures. In brief here, we write layer-by-layer, from bottom (deepest layers) to top, spot-by-spot with 2 μm steps (lateral and longitudinal) for possible merging between neighbor spots and so continuous writing. The pitch value is added on new fig. 2d. We also realize the energy conditions for these realizations were missing and have been also added. Finally, we mention in the manuscript the importance of the spherical aberration management without explaining the measures. We add also this information in the supplementary note.

Finally, we agree that our report is missing error bars or so-called “roughness” analyses important to support our reliable writing observations by quantitative data. We believe the most relevant uniformity analyses can be made on the basis of the repeated structures shown in original Fig. 1g (new Fig. 2c) and the phase plate demonstrations shown in Fig. 4-g (new Fig.6b). For the individual spots, we add the information about the spot size variability evaluated at <12% (relative SD). This information is added on page 6 of the manuscript (see marked version). For uniform writing, in the revised version, we give in the caption of Fig. 6b the evaluated standard deviations of the phase level for the processed squares. In the text, (see page 15 of marked version), we comment on these aspects. According to our measurements $\text{SD} < 0.1$ radians is achievable with simple point-by-point writing method used for these demonstrations. We discuss also the possible improvements by optimized scan writing and annealing methods.

W7- Dual-pulse ultrafast experiments using parallel and orthogonal polarization is reported (Shimotsuma, Y. et al. Appl. Phys. A 122, 159, 2016). The reported experimental concept using dual pulses seem to be a similar technique.

Obviously, we are not the first to apply a double-pulse approach for laser process optimizations. However, we remain convinced that our configuration to precisely rely on critical plasma seeds introduces a new concept that has never been exploited in any previous work. We believe there are two main reasons for this:

- First is the requirement of two different pulse characteristics so that we can individually control the plasma state up to the critical conditions and the energy deposition up to the writing level. This deviates from the reference given here relying on the use of two femtosecond pulses. In addition, it is interesting to note the very high energy levels used in this reference showing clearly experiments performed in totally different regimes. The modifications in this work were latter attributed to the limited temporal contrast of the laser technology used for these experiments (A. Wang et al., Phys. Rev. Research (2020)) and so conditions with, to our opinion, no clear relevance for comparison with our work.

- Second is a concept that is only immediately applicable in narrow-gap semiconductors as critical plasma conditions must be attainable without material modifications in order to exploit the plasma to control a separated writing beam. In particular, ultrafast laser 3D processing has been studied for decades and with many configurations in dielectrics. However, given the large bandgap values for dielectrics and shorter wavelength used in most experiments, the energy needed (absorbed) to create critical plasma conditions for the beams is above the material breakdown conditions. In the revised version of manuscript, we have made small adjustments (see marked version of the manuscript, page 17) of the last paragraph supporting a general double-pulse concept potentially

applicable to dielectrics provided that new schemes and in particular Mid-IR wavelengths are implemented.

W8- The statement of Line 125 is not convincing, claiming a “realization requiring a level of precision and flexibility out-of-reach for the most advanced single beam writing configurations”. In particular inside glass, there are numerous examples at similar precision and flexibility (e.g., Lei et al, *Optica*, 8, 11, 1365, 2021; Lancry et al, *Laser Photon Rev.* 7, 953, 2013; Delgoffe et al, *Optica*, 10, 10, 2023).

We agree that these and many other references could be cited to support a similar writing resolution in dielectrics. We acknowledge that the sentence does not clearly refer to the much more challenging problem of semiconductors. To clarify this point and solve this weakness, the sentence is reformulated by adding “...in semiconductors” and explicitly stating the resolution limitations with direct picosecond writing and the flexibility limitations for the other advanced solutions proposed to date (see marked version of the manuscript, page 7).

W9- The claim of achieving “the first clear evidence that amorphization in the bulk silicon” is confusing. Laser amorphization inside Si is reported in a number of articles and may be cited: Wang et al. *J. Laser Appl.* 36, 022015, 2024; Verburg et al., *Appl. Phys A.*, 120, 683, 2015).

We acknowledge an unclear formulation with “clear evidence”. We totally agree that amorphization in the bulk of silicon is regularly reported and that it is a reasonably expected transformation to explain the positive index variations observed in various laser writing experiments. However, attempts to reveal significant amorphization by advanced material analyses have been so far unsuccessful. Most TEM and/or electron diffraction analyses of modifications produced in nanosecond and picosecond regimes (including ours) counter-intuitively reveal in most cases an overall monocrystalline state. We agree that the provided references and (Trinh et al., *Advanced Engineering Materials* (2023) doi:10.1002/adem.202301377) are probably those that can support the best the formation of amorphous silicon. However, TEM images reveal relatively rare amorphous grains and Raman studies lead to an evaluation of the proportion of amorphous state not exceeding 0.2% in the best cases for the processed zones. According to the image shown in Fig. 3, we reveal a different regime leading to largely more amorphized grains in addition to organizational features for the grains never observed to date. To clarify, we have reformulated the mentioned sentence, largely expanding on this aspect by comparison with this reference (see marked version of the manuscript, pages 12 and 13).

Other comments:

1- What are the beam propagation directions in Fig. 2a, is this not critical for the model? The description in the caption of Fig. 2d does not match to the conceptual illustration in Fig. 2d. There is a phrase “pointing” in the figure, do the authors mean Poynting vector? Which color corresponds to which laser here? The associated plasma physics and its description is not clear in the text.

We have reworked the figure presentation to clarify/correct for beam directions and the other mentioned aspects, which are all critical. “Pointing” was for laser beam pointing to describe the lateral relative change of focus positions. This has been revised for clarity. We thank the reviewer and apology for the general lack of quality of the figures in the previous version.

2- Line 180 seems incorrect, $t_{fs}-t_{ps}$ = negative values would correspond to fs preionization. The discussion in Lines 185-187 (about pulse splitting and its interactions) is not clear.

We confirm the sign error. The negative delay corresponds to the fs preionization and the smallest features in Fig. 2g. The text and figure have been corrected. We have also improved the descriptions in lines 185-187 about the writing inhibition when the critical plasma seed is created during the writing pulse (see marked version of the manuscript, pages 9 and 10).

3- Are there any subsurface modifications that are not visible to the microscope images between the voxels effecting size measurements?

For voxel size measurements, we confirm all experiments were performed in a single plane (given depth at 300 μm) and well separated spots ($>50 \mu\text{m}$) to assure there is no distortion of the microscopy images caused by any other modifications.

4- Line 189: "Unprecedented high-degree of ionization" claim is given, but not clear with respect to what? The manuscript may benefit by refraining from ambiguous/unclear claims.

The plasma density level on which we rely, and which are required, have not been reported before. However, we agree that even if such conditions ($N \sim N_c$) have not been measured, similar laser conditions (1550nm, 200fs, NA=0.85) have already been applied to silicon. The main difference was that modifications were inaccessible in previous cases. Accordingly, we suppress the term "unprecedented" in the sentence to simply communicate we expect a "high-degree of ionization and the strongly nonequilibrium conditions" in the studied writing conditions (see marked version of the manuscript, page 10).

5- The following claim on reconfigurability may refer to previous work: "a new type of structures (see above) that permit near-perfect erasure by thermal annealing" Similar structures and thermal erasing is mentioned in page 21 of Chambonneau, Laser Photon Rev 15, 2100140, 2021.

We thank the reviewer for this relevant input. We agree the previously reported erasure capability by thermal treatment should not have been omitted as it surely represents a strong basis to support the reconfigurability aspects in our paper. In the revised manuscript (see marked version, page 13), we now refer clearly to this review paper (Chambonneau et al., Laser Photon Rev 15, 2100140, 2021) and the original report (Tokel et al., Nat. Photonics 11 639-645, 2017). With the changes, we can then emphasize even more on our advance with (i) the possibility to implement such thermal erasure using a laser for local treatment and (ii) the increased performance as previous works report typical few percent false-detection rates for information reading after erasure whereas we can demonstrate rewritable binary information without any false-detection over more than 100 cycles according to Fig 4e. This is a very important point for future reconfigurable technologies.

6- The anisotropy observed in Fig. 3d-e may be due to other effects such as birefringence. A more direct evidence would be required.

We agree that given the observations in Fig. 3h one may expect modifications exhibiting also form birefringence as for the nanogratings or nanopores more widely reported inside silica or other dielectrics. However, we rely in original Fig. 3d-e (now Fig. 4b,c) on lateral scattering imaging. This is not a transmission observation and a measurement of a polarization change through the structures. While birefringence must be also associated to the studied cases, there are sensitivity limitations for such a measurement. Having tentatively looked at the same structures with cross-polarized schemes

for birefringence detection, we have concluded on a much higher sensitivity with the scattering imaging approach. It was the only technical possibility to optically reveal an anisotropy for the structures written with low number of applied pulses. In the revised manuscript we add comments on the expected birefringence with the obtained anisotropy. We refer to the works having evidenced low-loss birefringent structures in dielectrics and our tentative birefringence measurements concluding on a very weak birefringence level in our studied cases in silicon (see marked version of the manuscript, pages 11 and 12).

7- The paper mentions spherical aberration compensation to maintain uniformity for writing at different depths. However, the discussion of depth control is quite limited.

We addressed this point in our response to W5-6. We apology for the missing description of the technical procedure for multi-level writing. The descriptions including the implemented spherical aberration compensations applied for the demonstrations are now added in section S2.2.

RESPONSE TO REVIEWER COMMENTS

Reviewer #1 (Remarks to the Author):

I am satisfied with the answers provided by the authors.

The work offers some novel contributions to the existing body of published papers on the topic. It is a broad interest for a large audience and will certainly stimulate further research on the topic, in particular on how to expand these methods towards other substrates.

We thank the reviewer for her/his time and comments on the quality and interest of our paper. We have been very pleased to address the previous comments which allowed us to improve the manuscript.

Reviewer #3 (Remarks to the Author):

The revised manuscript represents a meaningful technical advance in the field of ultrafast laser–material interactions, with direct relevance to semiconductor photonics. Following the revision, the central message of the paper is more clearly articulated and better supported by experimental evidence. I outline one major point that warrants further attention, along with several additional comments below.

We thank the reviewer the constructive comments during the evaluation. We are pleased to read this positive opinion on the new version of our paper. Below, we address the remaining points stressed in this report.

While the current manuscript offers a well-executed and experimentally rich contribution, it builds on earlier work by the authors (Phys. Rev. Res. 2, 033023, 2020), which had introduced the synergistic use of femtosecond and picosecond pulses to reduce modification thresholds in silicon. That study presents the foundational idea that an ultrafast pulse arriving shortly before another can prime the medium by injecting free carriers, enabling localized energy deposition. The present work extends this approach by intentionally engineering the plasma seeding conditions and demonstrates their utility in functional structuring. It would strengthen the manuscript if the authors could further elaborate on how the scope and mechanism of the present study build upon and differ from the earlier publication.

We thank the reviewer for this insightful comment. We fully agree that our previous paper (Phys. Rev. Res. 2, 033023, 2020) is one of the important works that influence the present advance. This previous paper was concentrating the temporal-contrast of the pulses as a critical parameter to provide the ability for internal structuring with ultrashort lasers. By intentionally modifying the contrast of applied pulses, it was found that a degraded contrast is actually leading to conditions more favorable for writing than perfectly contrasted femtosecond pulses which in more cases get delocalized during propagation in narrow gap materials. In particular, we can extrapolate from this work a decrease of writing threshold for picosecond pulses with a femtosecond pre-pulse component as is observed also in this work.

The current work, however, goes significantly beyond that foundational observation in several important ways. We would like to highlight that there was nothing in the previous work that could allow to predict the drastically improved writing performance (resolution and controllability) which

can be accessed with this scheme with properly engineered temporal components. In addition, apart from general considerations on potential pre-ionization seeding conditions with ultrashort pre-pulses, there was no evidence or considerations of the critical plasma conditions accessible under modification threshold inside semiconductors in the previous work. These aspects, we believe, represent a substantial advancement—both technically and conceptually—in understanding and applying spatio-temporal energy deposition using tailored ultrafast pulses.

While a detailed comparison with the previous work may risk overcomplicating the presentation, we fully agree with the reviewer on the importance of acknowledging this connection. Accordingly, we have added a citation when introducing the recent reports on the use of “... appropriate combinations of longer pulses.¹¹⁻¹⁵”. We also reformulate few sentences in page 5 so that we explicitly point the decrease of threshold observed in this and the previous work as we introduce the added new features with the configuration of this work (see page 5 of marked version of the manuscript).

The authors also demonstrate a range of functional outcomes: rewritable 3D optics, data storage, and tunable phase plates in silicon. The combination of rewriting capability and high repeatability represents a valuable advance. The integration of simulations, pump-probe microscopy, and application-oriented demonstrations provides support for the demonstrated approach.

We thank the reviewer for these positive comments on the scientific and technical value of our work.

Other comments:

- The QR codes shown in the manuscript do not appear to open or link successfully, returning a 404 error. This may be due to embedded images or limited resolution in the PDF version.

We have re-tested all QR codes images (bright field) and can confirm they are functional for regular QR scanners (presenting in front of the camera the images displayed on a computer monitor). This is functional with the images in the merged pdf file (manuscript) in the submission system and in supplementary information.

According to these tests, the 3 qr codes link to the urls:

<http://www.lp3.univ-mrs.fr/>

<https://ww.univ-amu.fr/en>

<https://www.cnrs.fr/en/cnrs>

The 404 error is a standard error to indicate that the browser was able to communicate with a given server, but the server could not find what was requested. We believe the reviewer got this error for the first url as it is now obsolete and replaced by www.lp3.fr. This must indicate the reading was successful but the webpage (an alias to the new page) was not functional at the time of the test.

We acknowledge that depending on the image quality and the scanner, it can be more or less difficult to read the codes but we have been able to successfully read the qr codes. Based in these considerations, we will pay a particular attention on the quality of the images in the final version of our paper, after editorial treatment.

- On page 15, the statement that “any phase value is achievable” is vague and potentially overstated—e.g., a full 2π phase shift is not explicitly demonstrated.

We agree that this statement is not totally appropriate and imprecise. We have replaced “... so that any phase value is achievable inside Si.” by “... so that a targeted phase value can be obtained inside Si.” (see p15 of marked version of the manuscript)

In the following sentence, the limitation in terms of maximum phase shift is stated: “...phase levels exceeding 1 radian are accessible for one processed plane...”

Similarly, for the statement on page 17 regarding “on-chip photonic memory devices never envisioned before”.

Monolithic silicon memories in the bulk of the materials is, to our knowledge, not accessible by any other approach. However, we agree that the formulation “... on-chip photonic memory devices never envisioned before by monolithic architectures.” Lead to an unnecessary emphasis on this aspect. In the revised manuscript, we simply suppress “never envisioned before” and we also replace “on-chip” by “in-chip” for a more precise statement. (see p17 of marked version of the manuscript)

- Are the 2-mm spherical-shaped samples used for a particular reason.

We acknowledge the paper was not explicitly justifying the sphere configuration. There is actually an important technical reason why we need to use of a spherical sample for lateral plasma diagnostics. We are discussing conditions for pulses focused at 0.85NA. The working distance of the lens used for most experiments in wafers (Olympus, LCPLN100XIR) is 1.2mm which does not allow to approach another high magnification objective (x20) to laterally observe the plasma formed below the wafer surface. For this reason, we decided to reproduce these excitation conditions at the center of a sphere. This leads to a solid-immersion scheme because we suppress the refraction of the Air-Si interface when the light is focused at the center. Accordingly, we can reproduce the conditions applied at 0.85NA inside the wafer by using a lens of numerical aperture given by $0.85/n_{Si}$ (that is less than 0.3) when working with the sphere. In practice, the lens at 0.3NA of the same objective series (Olympus LMPL10XIR) is used in this case to focus the laser. The working distance is 18mm which allows orthogonally setting another long working distance objective (Mitutoyo NIR, x20, WD 20mm) for lateral observation of the produced plasmas. The sphere configuration also enhances the imaging performance leading to an advantage for high-resolution observations. Details on the technical aspects of this configuration can be found in ref.11 (and its supplementary information).

We clarify these aspects by adding details at the end of the “methods” section (see marked version of the manuscript).